

# Decorated defect construction of gapless-SPT states

**Linhao Li[1], Masaki Oshikawa[1,2,3] and Yunqin Zheng[1,2]**

**1** Institute for Solid State Physics, University of Tokyo, Kashiwa, Chiba 277-8581, Japan
**2** Kavli Institute for the Physics and Mathematics of the Universe (WPI),
University of Tokyo, Kashiwa, Chiba 277-8583, Japan
**3** Trans-scale Quantum Science Institute, University of Tokyo,
Bunkyo-ku, Tokyo 113-0033, Japan

## Abstract

Symmetry protected topological (SPT) phases are one of the simplest, yet nontrivial, gapped systems that go beyond the Landau paradigm. In this work, we study an extension of the notion of SPT for gapless systems, namely, gapless symmetry protected topological states. We construct several simple gapless-SPT models using the decorated defect construction, which allow analytical understanding of non-trivial topological features including the symmetry charge under twisted boundary conditions, and boundary (quasi)-degeneracy under open boundary conditions. We also comment on the stability of the gapless-SPT models under symmetric perturbations, and apply small-scale exact diagonalization when direct analytic understanding is not available.

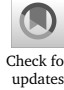
doi:10.21468/SciPostPhys.17.1.013

# 1 Introduction and summary

## 1.1 Gapped quantum matter

The study of topological phases of quantum matter has led to tremendous progress in understanding quantum many body systems beyond the Landau paradigm. The gapped phases are so far relatively well understood. Based on their symmetry and entanglement properties, the gapped phases can be classified into the following categories [1]:

1. **Trivially gapped phase:** There is a single ground state on an arbitrary spatial manifold, and a finite energy gap from the first excited state in the thermodynamic limit. The ground state preserves the global symmetry, and can be deformed to the trivial product state through *finite depth locally-symmetric unitary transformation* without closing the

energy gap. Its entanglement entropy obeys area law while the subleading contributions vanish in the thermodynamic limit. The ground state is short range entangled [1].

2. **Symmetry protected topological (SPT) phase:** Similarly to the trivially gapped phase, there is still a single ground state on an arbitrary *closed* spatial manifold and a finite energy gap from the first excited state in the thermodynamic limit. The ground state preserves the symmetry and is short-range entangled. The global symmetry should be anomaly free. However, unlike in the trivially gapped phase, when placing the system on a spatial manifold with nontrivial boundaries, due to the nontrivial physics appearing at the boundaries, there are either multiple ground states, or the energy spectrum becomes gapless in the thermodynamic limit. There is no finite depth locally-symmetric unitary transformation that maps the ground state to a trivial product state.[1] A systematic construction of gapped SPT phases is the decorated defect construction [7–9].

3. **Topological ordered (TO) phases and symmetry enriched topological (SET) phases:** The low energy is described by a symmetric topological quantum field theory (TQFT). The number of ground states depends on the topology of the spatial manifold. In particular when the spatial manifold is $S^d$ there is only one ground state. The ground states also have a finite energy gap from the first excited states in the thermodynamic limit. The entanglement entropy of the ground state has a constant contribution besides the area law part, which survives in the thermodynamic limit. This is termed topological entanglement entropy [10–13]. There are also nontrivial physics (e.g. gapless edge modes, spontaneous symmetry breaking or gapped TQFT) on the boundary when the spatial manifold is open. Finally, as the line operators (worldlines of anyons) are topological, they do not obey area law, and the theory is deconfined.

4. **Symmetry breaking phases:** There are multiple ground states even when the spatial manifold is $S^d$, due to spontaneous breaking of the global symmetry. These phases are within the Landau paradigm. There are also phases where the Landau symmetry breaking order and SPT/TO/SET orders coexist.

From the description above, it is clear that the SPT phase is the simplest, yet nontrivial, generalization of trivially gapped phase that goes beyond the Landau paradigm. We use *gapped SPT* phases to emphasize that the conventional SPT phases are for gapped systems.

## 1.2 Properties of gapless SPT states

In contrast to the gapped topological phases of quantum matter which are relatively well-understood, a systematic understanding of gapless quantum systems is still under development. See [14–26] for recent developments. The simplest type of gapless systems with nontrivial topological features are the so-called gapless symmetry protected topological states, studied in [14–16, 27, 28]. Let's summarize their common properties:

1. The gapless system has the global symmetry $\Gamma$. $\Gamma$ should be anomaly free and is not spontaneously broken by the ground state under *periodic* boundary conditions.

2. When placing the system on an arbitrary spatial manifold with *periodic* boundary conditions, the system should have a non-degenerate ground state with a finite size bulk gap which decays polynomially with respect to the system size.

---

[1]There are also exotic phases that do not require onsite unitary symmetries, but still satisfy the above properties, i.e. no degeneracy on closed manifolds and nontrivial boundary physics. They include Kitaev's $E_8$ state in $2+1$d [2,3] and $w_2 w_3$ theory in $4+1$d [4–6]. We also consider them as SPT phases where the symmetry is the spacetime diffeomorphism.

Table 1: Classification of gapless SPTs by whether they are purely gapless (horizontal direction) and intrinsically gapless (vertical direction).

|  | Contains gapped sector | No gapped sector |
|---|---|---|
| Non-intrinsic | gapless SPT [14, 15, 30] | purely gapless SPT [15] |
| Intrinsic | intrinsically gapless SPT [16, 30, 31] | intrinsically purely gapless SPT |

3. When placing the system on a spatial manifold with nontrivial boundaries, there are degenerate ground states with a finite size splitting decaying qualitatively faster (e.g. exponentially, or polynomially with a larger decaying constant) with respect to the system size.

4. When placing the system on a closed spatial manifold where the boundary conditions are twisted by the global symmetry $\Gamma$, a.k.a. twisted boundary conditions, the ground state carries nontrivial $\Gamma$ symmetry charge.

5. The criticality is confined. In particular, if the criticality has a 1-form symmetry, it should not be spontaneously broken.

The above properties are similar to those of the gapped SPT states, but there are major differences. For instance, the gap of gapless-SPT vanishes in the thermodynamical limit, while it remains open for the gapped-SPT. Moreover, the number of nearly degenerate states under OBC may differ from that of the gapped SPT. [2]

The examples of gapless-SPT states studied so far can be schematically organized by two features, as shown in Table 1.

- The vertical direction is distinguished by whether the gapless-SPT is intrinsic or non-intrinsic. If the topological features mentioned in the previous paragraph is can be realized by a gapped-SPT, then the gapless-SPT is non-intrinsic. Otherwise, it is intrinsic [16].

- The horizontal direction is distinguished by whether the gapless-SPT has a gapped sector. When there is a gapped sector, the degeneracy under OBC has at most exponential splitting decay. Otherwise, the splitting can be polynomial decaying, and was named as purely gapless-SPT.

The first example of gapless-SPT, which is non-intrinsic and contains a gapped sector, was first studied in [14]. The intrinsic gapless-SPT with a gapped sector was first studied in [16]. The gapless-SPT states without gapped sector (i.e. purely gapless SPT) was much less studied. The first example was found in [15] involving the time reversal symmetry and an on-site $\mathbb{Z}_2$ symmetry, and the terminology "purely" was proposed in [32], and examples with only on-site symmetries are demanding. A more systematic treatment of purely gapless-SPT states, both non-intrinsic and intrinsic, with on-site symmetries will be discussed in an upcoming

---

[2]We would like to comment that the fifth property is not implied by the first four. One example is the second order phase transition between a $(2+1)$d topological order and a trivially gapped phase. This system does not have any 0-form global symmetry and thus trivially satisfies the first four properties. Yet, as discussed in [29], this model has an emergent 1-form symmetry which is numerically demonstrated to be spontaneously broken, hence is deconfined. The fifth property is introduced to exclude this possibility.

For simplicity, we will use the following short hand notations to label the four classes of gapless-SPTs respectively:

- gSPT = gapless-SPT,

- igSPT = intrinsically gapless-SPT,

- pgSPT = purely gapless-SPT,

- ipgSPT = intrinsically purely gapless-SPT.

This work will focus on systems with a gapped sector, i.e. gSPT and igSPT.[3]

## 1.3 Decorated defect construction

A useful method to construct the gSPT and igSPT with a gapped sector is the decorated defect construction (DDC). The DDC was first used to construct gapped SPT states [7–9]. Applying the same construction to gapless system inspired the discovery of the first examples of gapless-SPT [14]. Later, by incorporating the symmetry extension method [34], the DDC also inspired the discovery of first examples of intrinsic gapless-SPT [16]. Our goal of this paper is to review this construction, and apply it to constructing bosonic spin models with on-site symmetries. Such models are simple, from which certain analytic results concerning their symmetry properties can be achieved. These models will also play an important role in our upcoming works [33, 35].

### 1.3.1 Constructing gapped SPT

The decorated defect construction was first devised to systematically construct gapped SPT phases, starting from the known lower dimensional gapped SPTs [7–9]. Suppose one would like to construct a gapped SPT system with global symmetry $\Gamma$. Assume $\Gamma$ fits into the symmetry extension

$$1 \to A \to \Gamma \to G \to 1, \tag{1}$$

where $A$ is the normal subgroup of $\Gamma$, and $G := \Gamma/A$. For simplicity, we assume that the extension is central, i.e. $G$ does not act on $A$.[4] One starts with a phase where $G$ is spontaneously broken, and on each codimension $p$ $G$-defect one decorates a $(d + 1 - p)$ dimensional gapped SPT protected by symmetry $A$ (i.e. $A$ gapped SPT). As we would like to eventually proliferate the $G$-defect network to restore the entire $\Gamma$ symmetry, the decorations should be consistent such that $G$-defect of each codimension should be free of $A$-anomaly, and in particular, there are no gapless modes localized on $G$-defects. Otherwise, if there are nontrivial gapless degrees of freedom localized on the $G$-defects, proliferation would not yield a gapped phase with one ground state. After defect proliferation, the resulting theory is a gapped SPT protected by the $\Gamma$ symmetry. The topological action of $\Gamma$ gapped SPT is given by the $\Gamma$ cocycle $\mathcal{F}^{\Gamma}_{d+1}$ which is a representative element in the cohomology group [36][5]

$$[\mathcal{F}^{\Gamma}_{d+1}] \in H^{d+1}(\Gamma, U(1)). \tag{2}$$

---

[3]Throughout this paper, "gSPT" specifically refers to non-intrinsic and not purely gapless-SPT. When we don't want to specify whether it is intrinsic or not, and would like to emphasize its gaplessness (to contrast with the gapped systems), we will use "gapless-SPT".

[4]The decorated defect construction of gapped SPTs was first discussed [7] in the special situation where the extension (1) is trivial, i.e. $\Gamma = A \times G$. The construction was later generalized to non-trivial extension (1) in [9].

[5]If $\Gamma$ is a continuous symmetry, the cohomology group should be $H^{d+1}(B\Gamma, U(1))$ where $B\Gamma$ is the classifying space of $\Gamma$.

We remark that a given $\Gamma$ can fit into multiple symmetry extensions with different pairs $(A, G)$. For a given extension $(A, G)$, as long as we exhaust all possible ways of decorating $A$ gapped SPT on $G$-defects, proliferating the $G$-defects exhausts all possible $\Gamma$ gapped SPTs. Hence different choices of $(A, G)$ yield the same set of $\Gamma$ gapped SPTs, and one can choose the most convenient pair $(A, G)$.

### 1.3.2 Constructing gapless-SPT

Let us proceed to construct the $\Gamma$ symmetric gapless-SPT states by modifying the decorated defect construction reviewed in section 1.3.1. We still assume that the global symmetry $\Gamma$ fits into the symmetry extension (1), and start with a gapped phase where $G$ is spontaneously broken. On each codimension $p$ $G$-defect, one decorates a $(d + 1 - p)$ dimensional $A$ gapped SPT. We finally fluctuate the $G$-defect network to the critical point, and define the critical point to be the gapless-SPT.

Comparing with the decorated defect construction of the gapped-SPT, the construction of the gapless-SPT has several important new features. As one no longer demands that fully proliferating the $G$-defect network leads to a gapped SPT phase, the consistency condition for the decoration can be relaxed. Depending on whether the consistency condition is preserved or relaxed, the resulting gapless-SPT are non-intrinsic and intrinsic respectively.

1. **gSPT:** The $A$ gapped SPTs decorated on the $G$-defects satisfy the same consistency condition as those for constructing the gapped SPT. Concretely, the $G$-defect of each codimension is free of $A$ anomaly. This means that further increasing the $G$-defect fluctuating strength leads to a $\Gamma$ gapped SPT, and gSPT is the phase transition between $G$ spontaneously broken phase and $\Gamma$ gapped SPT. In particular, when the extension (1) is trivial, i.e. $\Gamma = A \times G$, the construction was discussed in [14, 15]. See the left panel of figure 1 for the schematic phase diagram of gSPT.

2. **igSPT:** The $A$ gapped SPT decorated on the $G$-defects satisfies only a weaker, modified consistency condition. Concretely, the symmetry breaking phase we started with has a particular anomaly of a particular quotient group $\widehat{\Gamma}$ of $\Gamma$, where $G \subset \widehat{\Gamma}$. The choice of $\widehat{\Gamma}$ and its anomaly should be considered as part of input data of the construction. The defect decoration is constrained such that the anomaly of $\widehat{\Gamma}$ in the $G$ symmetry breaking phase is precisely cancelled against the anomaly induced by the defect decoration.[6] After decoration, the total symmetry group $\Gamma$ is anomaly free, and fluctuating the $G$-defect network to the critical point yields a $\Gamma$ anomaly free igSPT [16] . See the right panel of figure 1 for the schematic phase diagram of igSPT

It is natural to assume that the process of defect decoration and the process of $G$-defect fluctuation commute with each other. Then we may simplify the decorated defect construction by directly starting with a gapless critical system and decorating its $G$-defects. The gapless critical system is obtained by fluctuating the $G$-defects of the $G$ symmetry breaking phase before decorating the $A$ gapped SPTs, and from section 1.2 we require such critical system before decoration should have a non-degenerate ground state under periodic boundary condition, and is confined.[7] For the gSPT, we need to start with a critical point without any anomaly. While for the igSPT, we need to start with a critical point with a particular $\widehat{\Gamma}$ anomaly.

As commented in section 1.3.1, for a given $\Gamma$, there can be multiple choices of the symmetry extension (1). We noticed that the gapped SPT can be constructed using arbitrary $(A, G)$.

---

[6]The phenomenon of induced anomaly also appear in the discussion of anomalous-SPT [9, 37] and symmetry extended boundary of gapped SPT [34, 38].

[7]We will see in later sections that the defect decoration can be implemented by a unitary operation, which does not change the energy spectrum. This implies that the ground state degeneracy should be one both before and after defect decoration.

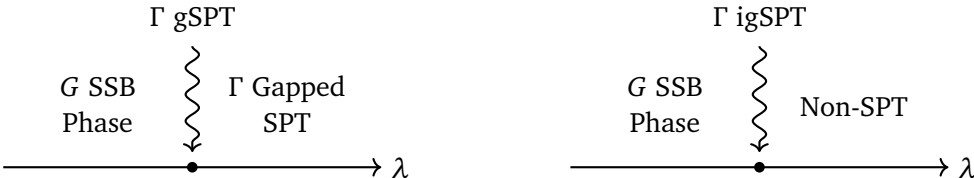

Figure 1: Phase diagram of non-intrinsically and intrinsically gapless-SPT. The horizontal axis is the strength of $G$-defect fluctuation. For the non-intrinsic case (left panel), the $G$-defects can be fully proliferated and one obtains $\Gamma$ gapped SPT. For the intrinsic case, one can only fluctuate the $G$-defects to the critical point. Further increase the fluctuation will not drive the system to $\Gamma$ symmetric gapped SPT phase.

However, this is no longer true for the igSPT. Note that one needs to specify an anomaly of $\widehat{\Gamma}$ (which includes $G$) as an input data of the decorated defect construction of igSPT. By definition, the resulting igSPT depends on the choice of symmetry extension (1), $\widehat{\Gamma}$ and the anomaly of $\widehat{\Gamma}$.

In this work, we will use the DDC to construct "canonical" bosonic spin models of gSPT and igSPT and discuss their topological properties. We also briefly comment on the stability under perturbations, while leaving an analytic study to an upcoming work [33].

## 1.4 Probing gSPT and igSPT

Given a gapless system with a non-degenerate ground state in the bulk with finite size, how can we tell whether it is a nontrivial gapless-SPT? If it is nontrivial, how can we tell whether it is intrinsic or non-intrinsic? There are several features commonly discussed in the literature:

1. degenerate ground states under OBC,

2. non-trivial symmetry charge of the ground state under the twisted boundary condition.

It is well-known that these features are useful in probing non-trivial gapped SPT phases [39–42]. The first feature is limited in two aspects: (1) It is useful for $(1 + 1)$d systems [39, 40], but for higher dimensions the boundary is extensive and the degeneracy on the boundary depends on the boundary dynamics. (2) For a generic Hamiltonian respecting the symmetry, the ground states on a finite open chain are only quasi-degenerate with exponentially small splittings, instead of being exactly degenerate [14]. This makes the identification of degenerate ground states subtle, especially in the gapless systems. While we can still separate the quasi-degenerate ground states with exponentially small finite-size excitation energies from gapless excitations with power-law finite-size excitation energies, the distinction can be challenging in practical numerical calculations.

We highlight that the second feature is merely based on the global symmetry, hence (1) can be applied to arbitrary spacetime dimension, and (2) is expected to be stable and exact for a generic Hamiltonian in the given gSPT and igSPT phase. This stability is also helpful for numerical calculations, as we will see later. See [43] for an application of twisted boundary condition to Lieb-Schultz-Mattis ingappability. Moreover, as discussed in [15], the charge under the twisted boundary condition is equivalent to the charge on the edge of the string order parameter for CFTs, and the latter is more commonly discussed in the literature. We prefer to discuss the twisted boundary condition rather than the string order parameter because the twisted boundary condition is less well-explored in the literature, and having a systematic and elementary discussion here should be more beneficial. Moreover, the twisted boundary condition can be generalized more naturally to higher dimensions.

### 1.5 Organization of the paper

We emphasize that the concepts and methods to be discussed in this paper, including the decorated defect construction, and the application of twisted boundary conditions to probe the gapless SPT, have been discussed in previous works already, in particular [14–16]. The goal of this paper is to apply the decorated defect construction to build concrete $(1 + 1)$d lattice spin models of gSPT and igSPT and study their properties in great detail. Our models are simple enough so that one can extract the ground state symmetry charges under various boundary conditions analytically, although the models are not exactly solvable.[8] Although the DDC was both applied to constructing gSPT in [14] and igSPT in [16], we believe that it is educational to present the construction of both gSPT and igSPT in a single place, highlighting the usefulness of DDC. It turns out that the examples constructed in this work pave the way to our later explorations of unified treatment of gSPT, igSPT, pgSPT and ipgSPT [33].

This paper is organized as follows. In section 2, we discuss in detail an analytically tractable example of gSPT, where $\Gamma = \mathbb{Z}_2 \times \mathbb{Z}_2, A = \mathbb{Z}_2, G = \mathbb{Z}_2$ and the spacetime dimension is $d = 1 + 1$. In section 3, we discuss in detail an analytically tractable example of igSPT, where $\Gamma = \mathbb{Z}_4, A = \mathbb{Z}_2, G = \widehat{\Gamma} = \mathbb{Z}_2$ and $d = 1 + 1$. We discuss a more realistic spin-1 model in section 4, which hosts both gSPT and igSPT simultaneously. There are several appendices. Appendix A shows the stability of boundary degeneracy of $\mathbb{Z}_2 \times \mathbb{Z}_2$ gapped SPT. Appendices B, C and D are devoted to further detailed discussions in section 3. Appendix E discusses an example of igSPT which involves time reversal symmetry. Appendix F shows the numerical result on the stability of igSPT under a certain symmetric perturbation.

## 2 gSPT: $(1 + 1)$d spin chains With $\mathbb{Z}_2 \times \mathbb{Z}_2$ symmetry

In this section, we study a concrete lattice model of gSPT: $(1 + 1)$d spin chain with global symmetry $\Gamma = \mathbb{Z}_2 \times \mathbb{Z}_2$. We let $A = \mathbb{Z}_2, G = \mathbb{Z}_2$, and the symmetry extension in (1) is trivial. For clarity, we use the superscript $A$ and $G$ to label the two $\mathbb{Z}_2$'s.

### 2.1 Spin chain construction

We construct the $1 + 1$d spin chain with $\Gamma = \mathbb{Z}_2^A \times \mathbb{Z}_2^G$ global symmetry. Since there are two $\mathbb{Z}_2$ symmetries, it is natural to assign two spin-$\frac{1}{2}$'s per unit cell: the spin-$\frac{1}{2}$'s living on the sites are charged under $\mathbb{Z}_2^G$ while those living in between the sites are charged under $\mathbb{Z}_2^A$. The symmetry operators are defined to be

$$U_A = \prod_{i=1}^{L} \tau_{i+\frac{1}{2}}^x, \qquad U_G = \prod_{i=1}^{L} \sigma_i^x, \tag{3}$$

where $\sigma_i^a$ and $\tau_{i+\frac{1}{2}}^a$, $a = x, y, z$, are Pauli matrices acting on the two spin-$\frac{1}{2}$'s, and $L$ is the number of unit cells. Both symmetry operators are on-site[9] and therefore $\Gamma$ is anomaly free. As explained in the introduction, we would like to start with a $\mathbb{Z}_2^G$ spontaneously broken phase, with the Hamiltonian

$$H_0 = -\sum_{i=1}^{L} \tau_{i+\frac{1}{2}}^x + \sigma_i^z \sigma_{i+1}^z. \tag{4}$$

---

[8]Our models do not involve fermions, but it can be shown that under Jordan-Wigner transformation, our model are equivalent to one of the models discussed in [44].

[9]A symmetry operator is on-site if it can be written as a product of local operators on mutually adjacent but un-overlapping patches, $U = \prod_i U_i$, where $i$ labels the patches.

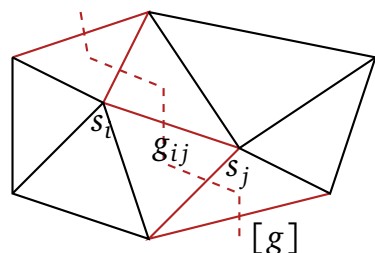

Figure 2: Triangulation of 2d spacetime. The black and red solid links are where the background field $g_{ij} = 0, 1$ respectively. The red dashed line in the dual lattice is the spacetime trajectory of the $\mathbb{Z}_2$ domain wall $[g]$, i.e. $\mathbb{Z}_2$ symmetry defect line. Flatness of $g$ ensures that $[g]$ forms loops.

It has two ground states

$$|\pm\rangle = \sum_{\{\tau^z_{i+\frac{1}{2}}\}} |\{\tau^z_{i+\frac{1}{2}}\}, \{\sigma^z_i = \pm 1\}\rangle. \tag{5}$$

Each of them spontaneously breaks $\mathbb{Z}_2^G$ but preserves $\mathbb{Z}_2^A$.

### 2.1.1 Domain wall decoration

To construct a $\mathbb{Z}_2^A \times \mathbb{Z}_2^G$ gSPT, we decorate each $\mathbb{Z}_2^G$ domain wall by a 0+1d $\mathbb{Z}_2^A$ SPT in a consistent way.[10] Each $\mathbb{Z}_2^G$ domain wall is associated with a $\mathbb{Z}_2^G$ group element $g$. $g = 0, 1$ means the domain wall is trivial/nontrivial, i.e. the adjacent $\sigma^z$ spin configurations are the same/opposite, respectively. We present the domain wall configuration using both the spacetime picture and the Hamiltonian picture.

**The spacetime picture:** It is useful to first discuss the domain wall in the spacetime picture. The spacetime is triangulated into 2-simplices. See figure 2 for an illustration. Each site $i$ is assigned a $\mathbb{Z}_2$ group element $s_i = 0, 1$, which corresponds to $\sigma^z_i = (-1)^{s_i}$ in the Hamiltonian picture. Each link is assigned a $\mathbb{Z}_2$ 1-cochain $g_{ij} = s_j - s_i$. The $g_{ij}$ is understood as a *flat* background field for the $\mathbb{Z}_2^G$ symmetry, and it measures the local domain wall excitation on the link. The locus where $g_{ij} = 1$ form a closed loop $[g]$ in the dual spacetime lattice, representing the worldline of the domain wall, a.k.a. the $\mathbb{Z}_2^G$ symmetry defect line. Decorating the $\mathbb{Z}_2^G$ domain wall by a 1d $\mathbb{Z}_2^A$ SPT [7, 14] means that we insert a $\mathbb{Z}_2^A$ Wilson line, a.k.a. 1d $\mathbb{Z}_2^A$ SPT, supported on $[g]$

$$\exp\left(i\pi \int_{[g]} a\right) = \exp\left(i\pi \int_{M_2} a \cup g\right), \tag{6}$$

in the path integral. The flatness of the $\mathbb{Z}_2^A$ background field $a$ ensures that the decoration is consistent: the domain wall junctions do not have $\mathbb{Z}_2^A$ anomaly. This fits into the construction of gSPT mentioned in section 1.3.2. The equality in (6) used the Poincare duality to transform the integral on $[g]$ into the integral over the entire 2d spacetime $M_2$. The topological term on the right hand side of (6) is precisely the effective action of $\mathbb{Z}_2^A \times \mathbb{Z}_2^G$ gapped SPT.

**The Hamiltonian picture:** In the Hamiltonian picture, domain wall decoration is implemented as follows [14]. We first identify the configuration representing the $\mathbb{Z}_2^G$ domain

---

[10]In $(1 + 1)$d, we only have codimension 1 defects, i.e. the domain walls. For this reason, the decorated defect construction is more conventionally called the decorated domain wall construction.

wall, i.e. $\sigma_i^z \sigma_{i+1}^z = -1$. Then on the link $(i, i+1)$, we stack a $\mathbb{Z}_2^A$ SPT (6), which assigns the wavefunction a minus sign if $\tau_{i+\frac{1}{2}}^z = -1$ (i.e. $a_{i,i+1} = 1$ in the spacetime picture) on the wall. Combining the two steps, one assigns a minus sign to the two configurations $(\sigma_i^z, \tau_{i+\frac{1}{2}}^z, \sigma_{i+1}^z) = (1, -1, -1), (-1, -1, 1)$ and leaves the wavefunction unchanged for other configurations. This operation can be realized by acting the unitary operator

$$U_{DW} = \prod_{i=1}^{L} \exp\left[\frac{\pi i}{4}(1-\sigma_i^z)\left(1-\tau_{i+\frac{1}{2}}^z\right)\right] \exp\left[\frac{\pi i}{4}(1-\sigma_{i+1}^z)\left(1-\tau_{i+\frac{1}{2}}^z\right)\right], \qquad (7)$$

on the states (5) [14]. In terms of the Hamiltonian, domain wall decoration just amounts to conjugating the original Hamiltonian (4) by $U_{DW}$, yielding

$$H_1 := U_{DW} H_0 U_{DW}^\dagger = -\sum_{i=1}^{L}\left(\sigma_i^z \tau_{i+\frac{1}{2}}^x \sigma_{i+1}^z + \sigma_i^z \sigma_{i+1}^z\right). \qquad (8)$$

The ground states of $H_1$ are still (5), but the first excited states associated with the domain wall excitations are decorated.

### 2.1.2 $\mathbb{Z}_2^A \times \mathbb{Z}_2^G$ gSPT

The next step is to fluctuate the decorated domain walls. It is helpful to discuss the fluctuation without decoration first. The fluctuation is well-known to be achieved by adding a transverse field $\Delta H = -\lambda \sum_{i=1}^{L} \sigma_i^x$, so that the $\mathbb{Z}_2^G$ spontaneously broken ferromagnetic phase of the Ising model (when $\lambda < 1$) is driven to the $\mathbb{Z}_2^G$ preserving paramagnetic phase (when $\lambda > 1$) where the domain walls are fully proliferated. The transition happens at $\lambda = 1$, which is of second order, and is described by a critical Ising CFT.

After domain wall decoration, the fluctuation should be realized by adding a *decorated* transverse field $U_{DW}\Delta H U_{DW}^\dagger = -\lambda \sum_{i=1}^{L} \tau_{i-\frac{1}{2}}^z \sigma_i^x \tau_{i+\frac{1}{2}}^z$. As the unitary transformation $U_{DW}$ does not change the energy spectrum, the critical point also takes place at $\lambda = 1$. The decorated model $U_{DW}(H_0 + \Delta H)U_{DW}^\dagger$ at $\lambda = 1$, is the $\mathbb{Z}_2^A \times \mathbb{Z}_2^G$ gSPT [14] (see also section 1.3.2 for the definition of gSPT)

$$H_{\text{gSPT}} = -\sum_{i=1}^{L}\left(\sigma_i^z \tau_{i+\frac{1}{2}}^x \sigma_{i+1}^z + \sigma_i^z \sigma_{i+1}^z + \tau_{i-\frac{1}{2}}^z \sigma_i^x \tau_{i+\frac{1}{2}}^z\right). \qquad (9)$$

When $\lambda > 1$, the domain wall is fully proliferated, yielding a $\mathbb{Z}_2^A \times \mathbb{Z}_2^G$ gapped SPT described by the well-known cluster model [45–47]

$$H_{\text{SPT}} = -\sum_{i=1}^{L}\left(\sigma_i^z \tau_{i+\frac{1}{2}}^x \sigma_{i+1}^z + \tau_{i-\frac{1}{2}}^z \sigma_i^x \tau_{i+\frac{1}{2}}^z\right). \qquad (10)$$

See figure 3 for the phase diagram before and after decoration.

As commented in section 1.3.2, we can simplify the above construction of gSPT by directly starting with the $\mathbb{Z}_2^G$ Ising CFT (whose Hamiltonian is given by $H_0 - \sum_{i=1}^{L} \sigma_i^x$), and conjugate it by $U_{DW}$. This simplification will be useful in section 3.

### 2.1.3 More on $U_{DW}$

We make a remark on the unitary operator $U_{DW}$. Although $H_{\text{gSPT}}$ and $H_0 - \sum_{i=1}^{L} \sigma_i^x$ are related through a unitary transformation $U_{DW}$, they are actually not equivalent as the $\mathbb{Z}_2^A \times \mathbb{Z}_2^G$ symmetric Hamiltonians. Recall that two $\Gamma$ symmetric Hamiltonians $H_1, H_2$ are considered

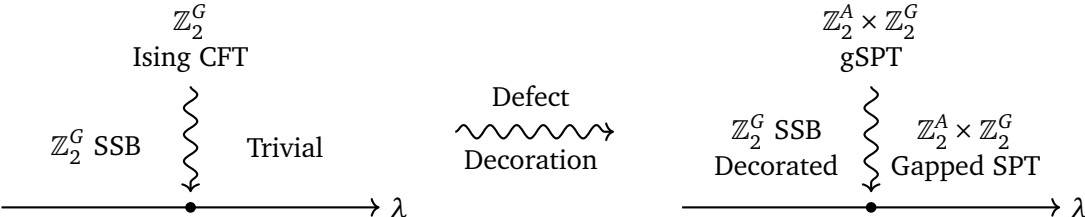

Figure 3: Phase diagram of $\mathbb{Z}_2^G$ Ising CFT (before decoration) and $\mathbb{Z}_2^A \times \mathbb{Z}_2^G$ gSPT (after decoration). The horizontal axis represents the transverse field $\lambda$.

equivalent if there is a *locally-symmetric* unitary transformation $U = \exp(i \int_{t_0}^{t_1} dt\, V(t))$ where $V(t)$ is a sum of local operators satisfying $[V(t), \Gamma] = 0$, such that $U H_1 U^\dagger = H_2$ [1,36]. Since $U_{DW}$ is a product of local unitary operators and each of them only acts on one or two unit cells, $U_{DW}$ is a local unitary transformation. Moreover, $U_{DW}$ on a closed chain with the periodic boundary condition is symmetric in the sense that $[U_{DW}, \Gamma] = 0$. Nevertheless, as each local operator $\exp\left(\frac{\pi i}{4}(1 - \sigma_i^z)\left(1 - \tau_{i+\frac{1}{2}}^z\right)\right)$ does not commute with $U_A$ and $U_G$, $U_{DW}$ is not a *locally-symmetric* unitary transformation. As an indication, $U_{DW}$ does not commute with the symmetry generator $\Gamma$ on an open chain, in contrast to the closed chain discussed above. In summary, $H_{\text{gSPT}}$ and $H_0 - \sum_{i=1}^{L} \sigma_i^x$ are *not* related by $\mathbb{Z}_2^A \times \mathbb{Z}_2^G$ *locally-symmetric* unitary transformation, hence they are not equivalent as $\mathbb{Z}_2^A \times \mathbb{Z}_2^G$ symmetric systems. This also justifies that the gSPT is *protected* by the $\mathbb{Z}_2^A \times \mathbb{Z}_2^G$.

It is interesting to compare $U_{DW}$ with the Kennedy-Tasaki (KT) transformation [48–50] introduced for integer-spin chains. Although the KT transformation is also implemented by a unitary operator $U_{KT}$, there are several differences. First, $U_{KT}$ is non-local, unlike $U_{DW}$ which is as discussed above a product of local unitary operators. Second, the KT transformation is useful for an open chain rather than for a closed chain, which is mapped to a non-local Hamiltonian by $U_{KT}$. Lastly, it maps a gapped SPT phase (on an open chain) to an SSB phase, while $U_{DW}$ maps a gapped SPT phase to a trivially gapped phase. The KT transformation will be relevant for the discussion in Section 4. In a later work by the same authors [35], we uncover that the KT transformation on a closed chain is related to the $U_{DW}$ in the following way, $\text{KT} = \text{KW} \cdot U_{DW} \cdot \text{KW}$, where KW is the Kramers-Wannier transformation for both $\mathbb{Z}_2^A \times \mathbb{Z}_2^G$ symmetries.

## 2.2 Trivializability upon stacking gapped SPTs

In this section, we will show that upon stacking a $\mathbb{Z}_2^A \times \mathbb{Z}_2^G$ gapped SPT, the $\mathbb{Z}_2^A \times \mathbb{Z}_2^G$ gSPT is equivalent to $\mathbb{Z}_2^G$ Ising criticality via a symmetric local unitary transformation.

Let us consider two decoupled systems. The first system is a $\mathbb{Z}_2^A \times \mathbb{Z}_2^G$ gSPT given by (9). The second system is a $\mathbb{Z}_2^A \times \mathbb{Z}_2^G$ gapped SPT given by (10). Since two systems are decoupled, the two Hamiltonians act on decoupled Hilbert spaces. We use the Pauli operators $\{\sigma_i^a, \tau_{i+\frac{1}{2}}^a\}$ for the first system, and $\{\widetilde{\sigma}_i^a, \widetilde{\tau}_{i+\frac{1}{2}}^a\}$ for the second system. The Hamiltonian for the entire system is the sum

$$H_{\text{gSPT}} + H_{\text{SPT}} = -\sum_{i=1}^{L} \left( \sigma_i^z \tau_{i+\frac{1}{2}}^x \sigma_{i+1}^z + \sigma_i^z \sigma_{i+1}^z + \tau_{i-\frac{1}{2}}^z \sigma_i^x \tau_{i+\frac{1}{2}}^z + \widetilde{\sigma}_i^z \widetilde{\tau}_{i+\frac{1}{2}}^x \widetilde{\sigma}_{i+1}^z + \widetilde{\tau}_{i-\frac{1}{2}}^z \widetilde{\sigma}_i^x \widetilde{\sigma}_{i+\frac{1}{2}}^z \right).$$

(11)

The decoupled system has enlarged global symmetry $(\mathbb{Z}_2^A \times \mathbb{Z}_2^G) \times (\widetilde{\mathbb{Z}}_2^A \times \widetilde{\mathbb{Z}}_2^G)$, whose generators

are

$$U_A = \prod_{i=1}^{L} \tau_{i+\frac{1}{2}}^x, \qquad U_G = \prod_{i=1}^{L} \sigma_i^x, \qquad \widetilde{U}_A = \prod_{i=1}^{L} \widetilde{\tau}_{i+\frac{1}{2}}^x, \qquad \widetilde{U}_G = \prod_{i=1}^{L} \widetilde{\sigma}_i^x. \tag{12}$$

There exists a symmetric local unitary transformation[11]

$$U_{\text{diag}} = \prod_{i=1}^{L} \exp\left(\frac{i\pi}{4}\left(1 - \sigma_i^z \tilde{\sigma}_{i+1}^z\right)\left(1 - \tau_{i+\frac{1}{2}}^z \tilde{\tau}_{i+\frac{1}{2}}^z\right)\right) \exp\left(\frac{i\pi}{4}\left(1 - \sigma_{i+1}^z \tilde{\sigma}_{i+1}^z\right)\left(1 - \tau_{i+\frac{1}{2}}^z \tilde{\tau}_{i+\frac{3}{2}}^z\right)\right), \tag{13}$$

which (locally) preserves the diagonal $\mathbb{Z}_2 \times \mathbb{Z}_2$, where two $\mathbb{Z}_2$'s are generated by $U_A \widetilde{U}_A$ and $U_G \widetilde{U}_G$ respectively. It is straightforward to check that

$$U_{\text{diag}}(H_{\text{gSPT}} + H_{\text{SPT}})U_{\text{diag}}^\dagger = -\sum_{i=1}^{L}\left(\tau_{i+\frac{1}{2}}^x + \sigma_i^z \sigma_{i+1}^z + \sigma_i^x + \widetilde{\tau}_{i+\frac{1}{2}}^x + \widetilde{\sigma}_i^x\right), \tag{14}$$

which is simply the Hamiltonian of the Ising CFT, a.k.a. the $\mathbb{Z}_2^G$ Landau transition, stacked with some trivially gapped degrees of freedom. In summary, we have shown that upon stacking a $\mathbb{Z}_2^A \times \mathbb{Z}_2^G$ gapped SPT, the $\mathbb{Z}_2^A \times \mathbb{Z}_2^G$ gSPT (9) is related to an ordinary $\mathbb{Z}_2^G$ Landau transition by a symmetric local unitary transformation. The above equivalence can be schematically represented as

$$\mathbb{Z}_2^A \times \mathbb{Z}_2^G \text{ gSPT } \oplus \mathbb{Z}_2^A \times \mathbb{Z}_2^G \text{ gapped SPT } \longleftrightarrow \mathbb{Z}_2^G \text{ Landau Transition}. \tag{15}$$

This implies that the nontrivial topological properties of the gSPT in the bulk (such as nontrivial charge of the ground state under the twisted boundary condition, see section 2.3.2) are basically inherited from the gapped SPT sector. However, we will find in section 2.3.3 that the boundary properties of the gSPT differ from those of the gapped SPT.

## 2.3 Symmetry features of $\mathbb{Z}_2^A \times \mathbb{Z}_2^G$ gSPT

We discuss the symmetry features of the $\mathbb{Z}_2^A \times \mathbb{Z}_2^G$ gSPT (9) that allow one to distinguish trivial vs nontrivial gSPTs. As motivated in the introduction (see section 1.4), we will consider the ground state degeneracy under open boundary condition (OBC), as well as the symmetry charge of the ground state under twisted boundary condition (TBC). We summarize the main properties in table 2.

### 2.3.1 Periodic boundary condition

On a finite chain with periodic boundary condition (PBC), the ground state of the $\mathbb{Z}_2^A \times \mathbb{Z}_2^G$ gSPT is non-degenerate. To see this, we first consider the Ising CFT described by the Hamiltonian $H_0 - \sum_{i=1}^{L} \sigma_i^x$. It is well-known that the critical Ising model has only one ground state on a finite chain, and the first excited state is separated from the ground state by a finite size gap decaying polynomially with respect to the system size. The non-degenerate ground state preserves the $\mathbb{Z}_2^A \times \mathbb{Z}_2^G$ global symmetry. Moreover, as noted in section 2.1.2, $H_{\text{gSPT}}$ and the Ising CFT have exactly the same energy eigenvalues because they are related via a unitary transformation $U_{DW}$, which implies that $H_{\text{gSPT}}$ also has a non-degenerate ground state on a finite closed chain, with a finite size gap, and is $\mathbb{Z}_2^G \times \mathbb{Z}_2^A$ symmetric under PBC.

---

[11]Without multiplying over $i$, each exponent in $U_{\text{diag}}$ commutes with the diagonal symmetries $U_A\widetilde{U}_A$ as well as $U_G\widetilde{U}_G$. As discussed in section 2.1.3, this implies that $U_{\text{diag}}$ is a *symmetric local unitary* transformation, which establishes the equivalence between different systems.

Table 2: Ground state degeneracy and symmetry charges of the ground state under PBC, TBC and OBC. $\mathbb{Z}_2^A$ (or $\mathbb{Z}_2^G$)-TBC means the boundary condition is twisted by $\mathbb{Z}_2^A$ (or $\mathbb{Z}_2^G$). We compare these properties between gSPT, Landau transition and gapped SPT, all with the same global symmetry $\mathbb{Z}_2^A \times \mathbb{Z}_2^G$. The $4 \to 2$ means that $H_{\text{gSPT}}$ has four ground states under OBC, but two of them are lifted under a symmetric perturbation localized on the boundary.

| | | $\mathbb{Z}_2^A \times \mathbb{Z}_2^G$ gSPT | $\mathbb{Z}_2^A \times \mathbb{Z}_2^G$ Landau Transition | $\mathbb{Z}_2^A \times \mathbb{Z}_2^G$ Gapped SPT |
|---|---|---|---|---|
| PBC: | GSD | 1 | 1 | 1 |
| | $\mathbb{Z}_2^A \times \mathbb{Z}_2^G$ Charge | $(0,0)$ | $(0,0)$ | $(0,0)$ |
| $\mathbb{Z}_2^A$-TBC: | GSD | 1 | 1 | 1 |
| | $\mathbb{Z}_2^A \times \mathbb{Z}_2^G$ Charge | $(0,1)$ | $(0,0)$ | $(0,1)$ |
| $\mathbb{Z}_2^G$-TBC: | GSD | 1 | 1 | 1 |
| | $\mathbb{Z}_2^A \times \mathbb{Z}_2^G$ Charge | $(1,0)$ | $(0,0)$ | $(1,0)$ |
| OBC: | GSD | $4 \to 2$ | 1 | 4 |

### 2.3.2 Twisted boundary condition

We show that on a closed chain with boundary condition twisted by $\mathbb{Z}_2^A$ (or $\mathbb{Z}_2^G$), the ground state of the $\mathbb{Z}_2^A \times \mathbb{Z}_2^G$ gSPT carries nontrivial symmetry charges under $\mathbb{Z}_2^G$ (or $\mathbb{Z}_2^A$) respectively. The same idea has been widely used to characterize nontrivial gapped SPT order [41,42,51–57], and here we use it to characterize the gSPT (and also igSPT in section 3).

**Twist by $\mathbb{Z}_2^A$:** We first twist the boundary condition using the $\mathbb{Z}_2^A$ symmetry (labeled by $\mathbb{Z}_2^A$-TBC), and measure the $\mathbb{Z}_2^G$ charge of the ground state. Twisting the boundary condition by $\mathbb{Z}_2^A$ means imposing a $\mathbb{Z}_2^A$ domain wall between sites $L - \frac{1}{2}$ and $L + \frac{1}{2}$ by changing the sign of the term $\tau_{L-\frac{1}{2}}^z \sigma_L^x \tau_{L+\frac{1}{2}}^z$. The gSPT Hamiltonian (9) becomes

$$H_{\text{gSPT}}^{\mathbb{Z}_2^A} = -\sum_{i=1}^{L-1}\left(\sigma_i^z \tau_{i+\frac{1}{2}}^x \sigma_{i+1}^z + \sigma_i^z \sigma_{i+1}^z + \tau_{i-\frac{1}{2}}^z \sigma_i^x \tau_{i+\frac{1}{2}}^z\right) - \sigma_L^z \tau_{L+\frac{1}{2}}^x \sigma_1^z - \sigma_L^z \sigma_1^z + \tau_{L-\frac{1}{2}}^z \sigma_L^x \tau_{L+\frac{1}{2}}^z. \tag{16}$$

It is useful to note that the twisted and untwisted gSPT Hamiltonian are related by a unitary transformation $H_{\text{gSPT}}^{\mathbb{Z}_2^A} = \sigma_L^z H_{\text{gSPT}} \sigma_L^z$, hence the ground state of $H_{\text{gSPT}}^{\mathbb{Z}_2^A}$ is also non-degenerate. Denote the ground state under PBC as $|\text{GS}\rangle$, and that under $\mathbb{Z}_2^A$-TBC as $|\text{GS}\rangle_{\text{tw}}^{\mathbb{Z}_2^A}$. We have

$$|\text{GS}\rangle_{\text{tw}}^{\mathbb{Z}_2^A} = \sigma_L^z |\text{GS}\rangle . \tag{17}$$

It follows that

$$U_G |\text{GS}\rangle_{\text{tw}}^{\mathbb{Z}_2^A} = U_G \sigma_L^z U_G^\dagger U_G |\text{GS}\rangle = -\sigma_L^z |\text{GS}\rangle = -|\text{GS}\rangle_{\text{tw}}^{\mathbb{Z}_2^A} , \tag{18}$$

which shows that $|\text{GS}\rangle_{\text{tw}}^{\mathbb{Z}_2^G}$ has $\mathbb{Z}_2^G$ charge 1.[12]

---

[12]We used the fact that the ground state under PBC is neutral under $\mathbb{Z}_2^G$. More precisely, (18) only shows the relative charge, i.e. the $\mathbb{Z}_2^G$ charge of the ground state under TBC minus that under PBC, is one. The relative charge will be useful in section 3.

**Twist by $\mathbb{Z}_2^G$:** We can alternatively twist the boundary condition using $\mathbb{Z}_2^G$ symmetry (labeled by $\mathbb{Z}_2^G$-TBC), and measure the $\mathbb{Z}_2^A$ charge of the ground state. Twisting the boundary condition by $\mathbb{Z}_2^G$ means imposing a $\mathbb{Z}_2^G$ domain wall on the link between $L$-th and 1st sites, by changing the sign of the terms $\sigma_L^z \sigma_1^z$ and $\sigma_L^z \tau_{L+\frac{1}{2}}^x \sigma_1^z$. The gSPT Hamiltonian (9) becomes

$$H_{\text{gSPT}}^{\mathbb{Z}_2^G} = -\sum_{i=1}^{L-1} \left( \sigma_i^z \tau_{i+\frac{1}{2}}^x \sigma_{i+1}^z + \sigma_i^z \sigma_{i+1}^z + \tau_{i-\frac{1}{2}}^z \sigma_i^x \tau_{i+\frac{1}{2}}^z \right) + \sigma_L^z \tau_{L+\frac{1}{2}}^x \sigma_1^z + \sigma_L^z \sigma_1^z - \tau_{L-\frac{1}{2}}^z \sigma_L^x \tau_{L+\frac{1}{2}}^z .$$

(19)

Note that $\sigma_i^z \tau_{i+\frac{1}{2}}^x \sigma_{i+1}^z$ commutes with every term in $H_{\text{gSPT}}^{\mathbb{Z}_2^G}$, the ground state $|\text{GS}\rangle_{\text{tw}}^{\mathbb{Z}_2^G}$ should be its eigen-vector

$$\sigma_i^z \tau_{i+\frac{1}{2}}^x \sigma_{i+1}^z |\text{GS}\rangle_{\text{tw}}^{\mathbb{Z}_2^G} = U_{DW}^\dagger \tau_{i+\frac{1}{2}}^x U_{DW} |\text{GS}\rangle_{\text{tw}}^{\mathbb{Z}_2^G} = \begin{cases} |\text{GS}\rangle_{\text{tw}}^{\mathbb{Z}_2^G}, & i = 1, \ldots, L-1, \\ -|\text{GS}\rangle_{\text{tw}}^{\mathbb{Z}_2^G}, & i = L. \end{cases}$$

(20)

Consequently, the ground state has $\mathbb{Z}_2^A$ charge 1:

$$U_A |\text{GS}\rangle_{\text{tw}}^{\mathbb{Z}_2^G} = \prod_{i=1}^{L} \tau_{i+\frac{1}{2}}^x |\text{GS}\rangle_{\text{tw}}^{\mathbb{Z}_2^G} = -\prod_{i=1}^{L} (\sigma_i^z \sigma_{i+1}^z) |\text{GS}\rangle_{\text{tw}}^{\mathbb{Z}_2^G} = -|\text{GS}\rangle_{\text{tw}}^{\mathbb{Z}_2^G} .$$

(21)

In summary, we find that when we use $\mathbb{Z}_2^{A,G}$ to twist the boundary condition on a closed chain, the ground state of the twisted Hamiltonian has nontrivial $\mathbb{Z}_2^{G,A}$ charge. This is the property distinguished from the $\mathbb{Z}_2^A \times \mathbb{Z}_2^G$ Landau transition, where its ground state under the twisted boundary conditions does not carry any nontrivial symmetry charge. This tells us that we can use the symmetry charge of the ground state in the twisted sector as a topological invariant to distinguish the nontrivial gSPT from trivial gSPT (e.g. second order Landau transition). On the other hand, the symmetry charges under TBC coincide with those of the gapped SPT. We summarize the results in table 2.

### 2.3.3 Open boundary condition

As the nontrivial boundary modes protected by the global symmetry is a signature of gapped SPT, we will find that same is true for the gSPT. We use the symmetry to analytically show that the ground states of $H_{\text{gSPT}}$ have to be exactly degenerate under OBC, but the number of degeneracy differs from the gapped SPT. This phenomenon was discussed in [14, 15].

We place the spin system on an open chain. The left most spin is the $\sigma$ spin, and the right most spin is the $\tau$ spin. The $\sigma$ spins are supported on $i = 1, \ldots, L$, and the $\tau$ spins are supported on $i + \frac{1}{2} = \frac{3}{2}, \ldots, L + \frac{1}{2}$. We first choose the OBC such that only the interactions completely supported on the chain are kept. The Hamiltonian is

$$H_{\text{gSPT}}^{\text{OBC}} = -\sum_{i=1}^{L-1} \left( \sigma_i^z \tau_{i+\frac{1}{2}}^x \sigma_{i+1}^z + \sigma_i^z \sigma_{i+1}^z \right) - \sum_{i=2}^{L} \tau_{i-\frac{1}{2}}^z \sigma_i^x \tau_{i+\frac{1}{2}}^z ,$$

(22)

and the symmetry operators are

$$U_A = \prod_{i=1}^{L} \tau_{i+\frac{1}{2}}^x , \qquad U_G = \prod_{i=1}^{L} \sigma_i^x .$$

(23)

We find that the set of operators $\{\sigma_1^z, \tau_{L+\frac{1}{2}}^z, \sigma_L^z \tau_{L+\frac{1}{2}}^x, U_A, U_G\}$ all commute with the Hamiltonian, hence the ground state degeneracy must be at least the dimension of its irreducible

representation. To find the representation, we choose the maximally commuting subset of operators as $\{\sigma_1^z, \tau_{L+\frac{1}{2}}^z\}$, and denote their eigenvalue of a particular ground state $|\psi\rangle$ by $(a, b)$, where $a, b = \pm 1$. It is then possible to generate other ground states with different quantum numbers as follows:

$$
\begin{array}{c|c|c}
 & \sigma_1^z & \tau_{L+\frac{1}{2}}^z \\
\hline
|\psi\rangle & a & b \\
\hline
U_A |\psi\rangle & a & -b \\
\hline
U_G |\psi\rangle & -a & b \\
\hline
U_A U_G |\psi\rangle & -a & -b
\end{array}
\tag{24}
$$

This shows that there must be at least four exactly degenerate ground states of $H_{\text{gSPT}}^{\text{OBC}}$ of four different sets of quantum numbers. Numerical exact diagonalization confirms that the ground state degeneracy is exactly four.

However, symmetry does not forbid us to perturb (22) by adding symmetric boundary terms. We can add a boundary interaction

$$
\Delta H_{\text{gSPT}}^{\text{OBC}} = -\tau_{L+\frac{1}{2}}^x,
\tag{25}
$$

which changes the original OBC to a new OBC. This interaction does not commute with $\tau_{L+\frac{1}{2}}^z$, so the set of operators commuting with the Hamiltonian $H_{\text{gSPT}}^{\text{OBC}} + \Delta H_{\text{gSPT}}^{\text{OBC}}$ reduces to $\{\sigma_1^z, \sigma_L^z \tau_{L+\frac{1}{2}}^x, U_A, U_G\}$. As a consequence, the dimension of irreducible representation reduces from four to two. Indeed, numerical exact diagonalization confirms that there are only two exactly degenerate ground states under the new OBC. This degeneracy splitting was already noted in [14, 15]. Here, we provide a simple analytical argument of this splitting by finding the representation. In appendix A, we show that arbitrary finite range perturbation does not lift the 4-fold exact degeneracy of the $\mathbb{Z}_2^A \times \mathbb{Z}_2^G$ gapped SPT.

## 2.4 Instability of gSPT

As noted in table 2, the symmetry properties of the ground states under the twisted boundary condition are the same for the gSPT and gapped SPT. Is there a symmetric perturbation of the gSPT which derives gSPT to the gapped SPT? In this subsection, we confirm this by noting that such a $\mathbb{Z}_2^A \times \mathbb{Z}_2^G$ symmetric perturbation exists, which is

$$
V = -h \sum_{i=1}^{L} \tau_{i-\frac{1}{2}}^z \sigma_i^x \tau_{i+\frac{1}{2}}^z, \quad h > 0.
\tag{26}
$$

In other words, adding $V$ to $H_{\text{gSPT}}$ simply modifies the coefficient of the last term of (9) by $-h$. After undoing the domain wall decoration by conjugating the $H_{\text{gSPT}} + V$ by $U_{DW}$, we get

$$
U_{DW}(H_{\text{gSPT}} + V)U_{DW}^\dagger = -\sum_{i=1}^{L} \left( \tau_{i+\frac{1}{2}}^x + \sigma_i^z \sigma_{i+1}^z + (1+h)\sigma_i^x \right),
\tag{27}
$$

which is just a critical Ising model perturbed by $-h \sum_i \sigma_i^x$. It is well-known that under fermionization, this term is the mass term of the Majorana fermion, which is a relevant perturbation. This shows that adding a perturbation with infinitesimal $h$ drives the gSPT to gapped SPT phase, which shows that gSPT is unstable under symmetric perturbation towards gapped SPT phases.

# 3 igSPT: $(1+1)$d spin chain with $\mathbb{Z}_4$ symmetry

In this section, we study a concrete lattice model of igSPT: $(1+1)$d spin chain with $\mathbb{Z}_4$ global symmetry. We let $A = \mathbb{Z}_2$, $G = \mathbb{Z}_2$, and the symmetry extension (1) is now nontrivial. We still use superscripts $A$ and $G$ to label the two $\mathbb{Z}_2$'s, and use superscript $\Gamma$ to label $\mathbb{Z}_4$.

The igSPT was first studied in [16]. Although essentially all the features of igSPT have been discussed in that work, the model discussed in our present work has the advantage of being simpler, where many symmetry properties can be extracted exactly without numerical computation. Moreover, the characterization of topological features in [16] heavily uses the string order parameter, while in our work we provide some alternative perspectives of characterization using the twisted boundary conditions. Although the charge of the string order parameter and the charge of ground states under the TBC are known to be related in the continuum limit [15], it is nevertheless beneficial to discuss the TBC on the lattice and compare with the string order parameter discussion on the lattice in [16].

## 3.1 Spin chain construction

### 3.1.1 Domain wall decoration and induced anomaly

**Domain wall decoration:**   We construct the $(1+1)$d spin chain with $\mathbb{Z}_4^\Gamma$ global symmetry, by applying the decorated defect construction reviewed in section 1.3.2. Concretely, we start with a $\mathbb{Z}_2^G$ symmetry spontaneously broken phase with a nontrivial anomaly of $\mathbb{Z}_2^G$, and then decorate the $\mathbb{Z}_2^G$ domain wall by $\mathbb{Z}_2^A$ SPT. We will show below that the domain wall decoration induces a nontrivial $\mathbb{Z}_2^G$ anomaly due to the nontrivial extension (1), and two $\mathbb{Z}_2^G$ anomalies are designed to cancel against each other. Thus the entire $\mathbb{Z}_4^\Gamma$ symmetry is anomaly free. We further proliferate the decorated $\mathbb{Z}_2^G$ domain wall, and fine tune the system to the critical point. The resulting critical point is the $\mathbb{Z}_4^\Gamma$ igSPT.

**Induced anomaly:**   We explain why the domain wall decoration induces nontrivial $\mathbb{Z}_2^G$ anomaly. Let us denote the background fields of $\mathbb{Z}_2^G$ and $\mathbb{Z}_2^A$ as $g$ and $a$ respectively, both of which are 1-cochains. The $\mathbb{Z}_4^\Gamma$ background field is $2a - \tilde{g}$, where $\tilde{g}$ is a lift of $g$ to a $\mathbb{Z}_4^\Gamma$ valued cochain, i.e. $g = \tilde{g} \mod 2$. By requiring the $\mathbb{Z}_4^\Gamma$ background to be flat, we find

$$\delta(2a - \tilde{g}) = 2\delta a - \delta\tilde{g} = 0 \mod 4, \tag{28}$$

which implies

$$\delta a = \text{Bock}(g) := \frac{1}{2}\delta\tilde{g} \mod 2, \qquad \delta g = 0 \mod 2. \tag{29}$$

$\text{Bock}(g)$ is the Bockstein of $g$, which is defined as in (29). As (6), decorating the $\mathbb{Z}_2^G$ domain wall by a 1d $\mathbb{Z}_2^A$ SPT means stacking a $\mathbb{Z}_2^A$ Wilson line to the worldline of $\mathbb{Z}_2^G$ domain wall. However, due to the nontrivial bundle constraint (29), the domain wall decoration is not gauge invariant, and equivalently it induces a nontrivial dependence on the extension to the 3d bulk $M_3$,

$$\exp\left(i\pi \int_{[g]} a\right) = \exp\left(i\pi \int_{M_2} a \cup g\right) = \exp\left(i\pi \int_{M_3} g \cup \text{Bock}(g)\right). \tag{30}$$

In the second equality, we applied total derivative to promote the 2d integral to the 3d integral and used (29). A physical interpretation of (30) is that domain wall decoration induces a $\mathbb{Z}_2^G$ anomaly. We will denote this anomaly as the *induced anomaly*.

However, the igSPT by definition should be free of $\mathbb{Z}_4^\Gamma$ anomaly, and the system should be independent of the extension to $M_3$. This demands that the $\mathbb{Z}_2^G$ spontaneously broken system

before domain wall decoration should already exhibit an opposite anomaly of $\mathbb{Z}_2^G$, which is given by the same inflow action

$$\exp\left( i\pi \int_{M_3} g \cup \text{Bock}(g) \right). \tag{31}$$

After domain wall decoration, the anomaly (31) from the low energy cancels against the induced anomaly (30) from the domain wall decoration, and the total system is anomaly free.

As commented at the end of section 2.1.2, one can simplify the discussion by directly starting with a critical system with a non-degenerate ground state and a $\mathbb{Z}_2^G$ anomaly (31). A standard candidate is the critical boundary theory of $(2+1)$d $\mathbb{Z}_2^G$ SPT, known as the Levin-Gu model [42]. We then decorate the $\mathbb{Z}_2^G$ domain walls (via conjugating by the unitary operator $U_{DW}$ in (7)). We will take this simplified strategy of domain wall decoration below.

### 3.1.2 The model

We still let the $\sigma$ spins live on integer sites and $\tau$ spins live on half integer sites. Let us start from the Levin-Gu model [42]

$$H_{\text{LG}} = -\sum_{i=1}^{L} \left( \sigma_i^x - \sigma_{i-1}^z \sigma_i^x \sigma_{i+1}^z \right), \tag{32}$$

with an anomalous $\mathbb{Z}_2^G$ symmetry transformation:

$$U_G = \prod_{i=1}^{L} \sigma_i^x \prod_{i=1}^{L} \exp\left( \frac{i\pi}{4} (1 - \sigma_i^z \sigma_{i+1}^z) \right). \tag{33}$$

The $\mathbb{Z}_2^G$ symmetry operator is realized in a non-on-site way, which is demanded by the $\mathbb{Z}_2^G$ anomaly.

In the next step, we consider the following Hamiltonian which couples the $\tau$ and $\sigma$ spins and serves as the pre-decorated Hamiltonian:

$$H_{\text{pre}} = -\sum_{i=1}^{L} \left( \sigma_i^x - \sigma_{i-1}^z \tau_{i-\frac{1}{2}}^x \sigma_i^x \tau_{i+\frac{1}{2}}^x \sigma_{i+1}^z + \tau_{i-\frac{1}{2}}^x \right). \tag{34}$$

This Hamiltonian is invariant under the $\mathbb{Z}_4$ symmetry transformation:

$$U_\Gamma^{\text{pre}} = \prod_{i=1}^{L} \sigma_i^x \prod_{i=1}^{L} \exp\left( \frac{i\pi}{4} \left( 1 - \sigma_i^z \tau_{i+\frac{1}{2}}^x \sigma_{i+1}^z \right) \right). \tag{35}$$

The normal subgroup $\mathbb{Z}_2^A$ is generated by an on-site operator

$$U_A = (U_\Gamma^{\text{pre}})^2 = \prod_{i=1}^{L} \tau_{i+\frac{1}{2}}^x. \tag{36}$$

Indeed, the two Hamiltonians (34) and (32) enjoy the same low energy theory. As the last term in (34) commutes with the rest of the terms, the ground state should be the eigenstate of $\tau_{i-\frac{1}{2}}^x$ with eigenvalue 1. See appendix C for a more detailed discussion on this point.[13] In the low energy sector, we simply substitute $\tau_{i-\frac{1}{2}}^x = 1$ in (34), and obtain that the low

---

[13]In fact, all the low energy states with energy $E - E_{\text{GS}} \ll 1$ satisfy $\tau_{i-\frac{1}{2}}^x = 1$.

energy effective Hamiltonian (34) is precisely the Levin-Gu Hamiltonian (32). Moreover, the $\mathbb{Z}_2^A$ normal subgroup decouples from the low energy. Only $\mathbb{Z}_2^G$ acts nontrivially on the low energy degrees of freedom, in the same way as (33):

$$U_\Gamma^{\text{pre}}|_{\text{low}} = \prod_{i=1}^{L} \sigma_i^x \prod_{i=1}^{L} \exp\left( \frac{i\pi}{4}(1 - \sigma_i^z \sigma_{i+1}^z) \right). \tag{37}$$

The additional $\tau$ operators as in (34) and (35) are motivated by the group extension. We would like to introduce $\tau$ operators such that $U_A = \prod_i \tau_{i+\frac{1}{2}}^x$ generates an anomaly free $\mathbb{Z}_2^A$, extending the $\mathbb{Z}_2^G$ to $\mathbb{Z}_4^\Gamma$. In other words, we demand a modification of $U_G$ in (33) such that it squares to $U_A$. This is precisely achieved by replacing $\sigma_i^z \sigma_{i+1}^z$ by $\sigma_i^z \tau_{i+\frac{1}{2}}^x \sigma_{i+1}^z$. This further induces how the Levin-Gu Hamiltonian (32) should be mapped to (34).

Then, the Hamiltonian for the $\mathbb{Z}_4^\Gamma$ igSPT is obtained by conjugating (34) using the unitary operator $U_{DW}$. The Hamiltonian is

$$H_{\text{igSPT}} = U_{DW} H_{\text{pre}} U_{DW}^\dagger = -\sum_{i=1}^{L} \left( \tau_{i-\frac{1}{2}}^z \sigma_i^x \tau_{i+\frac{1}{2}}^z + \tau_{i-\frac{1}{2}}^y \sigma_i^x \tau_{i+\frac{1}{2}}^y + \sigma_{i-1}^z \tau_{i-\frac{1}{2}}^x \sigma_i^z \right). \tag{38}$$

The pre-decorated $\mathbb{Z}_4^\Gamma$ symmetry operator becomes

$$U_\Gamma = U_{DW} U_\Gamma^{\text{pre}} U_{DW}^\dagger = \prod_{i=1}^{L} \sigma_i^x \prod_{i=1}^{L} \exp\left( \frac{i\pi}{4}\left( 1 - \tau_{i+\frac{1}{2}}^x \right) \right), \tag{39}$$

under which $\sigma_i^x \to \sigma_i^x, \sigma_i^{y,z} \to -\sigma_i^{y,z}$, $\tau_{i+\frac{1}{2}}^x \to \tau_{i+\frac{1}{2}}^x$, $\tau_{i+\frac{1}{2}}^y \to \tau_{i+\frac{1}{2}}^z$, $\tau_{i+\frac{1}{2}}^z \to -\tau_{i+\frac{1}{2}}^y$. $\mathbb{Z}_4^\Gamma$ is anomaly free, which can be seen from the on-site-ness of $U_\Gamma$ [14] and justifies that $\mathbb{Z}_2^G$ anomaly in pre-decorated Hamiltonian is canceled by the induced anomaly of decorated defect construction. The normal subgroup $\mathbb{Z}_2^A$ is also generated by an on-site operator Eq. (36). Here we remark that the Hamiltonian (38) is Jordan-Wigner dual to Eq.49 and Eq.50 in [44].

## 3.2 Symmetry features of $\mathbb{Z}_4^\Gamma$ igSPT

We discuss the symmetry features of the $\mathbb{Z}_4^\Gamma$ igSPT (38). An immediate fact to realize is that there is no $\mathbb{Z}_4^\Gamma$ gapped SPT in $(1+1)$d.[15] Thus it is not possible to stack a gapped SPT to unitarily connect it to another possibly more trivial igSPT. For this reason, the origin of the nontrivial SPT order at the critical point here is less obvious, in contrast to the $\mathbb{Z}_2^A \times \mathbb{Z}_2^G$ gSPT. In this subsection, we discuss its properties under various boundary conditions. We summarize the main results in table 3.

### 3.2.1 Periodic boundary condition

We have motivated in section 1.2 that any igSPT should have one non-degenerate ground state, with a finite size splitting with the first excited state. Thus we would like to check the ground state degeneracy of (38) under PBC to be one.

As we find in section 3.1.2, the number of ground states of the $\mathbb{Z}_4^\Gamma$ igSPT is identical to that of the Levin-Gu model (32). In appendix B.1, we show, by Jordan-Wigner transformation, that the number of ground states of the Levin-Gu model depends on $L \mod 4$ and is given as

$$\text{GSD}_L = \begin{cases} 2, & L = 2 \mod 4, \\ 1, & \text{otherwise.} \end{cases} \tag{40}$$

---

[14]However, not every anomaly free symmetry operator is on-site.

[15]The $(1+1)$d bosonic SPT with a discrete symmetry $G$ is classified by $H^2(G, U(1))$. In our case, $G = \mathbb{Z}_4$, and it is well-known [36] that $H^2(\mathbb{Z}_4, U(1)) = 0$ is trivial, hence there is no nontrivial $\mathbb{Z}_4$ SPT phase in $(1+1)$d.

Table 3: Ground state degeneracy and symmetry charges of the ground state under PBC, TBC and OBC. We focus on the system size $L = 0, 1, 3, 4, 5, 7 \mod 8$ to ensure trivial ground state degeneracy. Relative $\mathbb{Z}_2^A$ (or $\mathbb{Z}_4^\Gamma$) charge means the difference between the corresponding charge under the TBC and that under the PBC. We compare these properties between the igSPT and Landau transition, both with the same global symmetry $\mathbb{Z}_4^\Gamma$.

| | | $\mathbb{Z}_4^\Gamma$ igSPT | $\mathbb{Z}_4^\Gamma$ Landau Transition |
|---|---|---|---|
| PBC: | GSD | 1 | 1 |
| $\mathbb{Z}_2^A$-TBC: | GSD | 1 | 1 |
| | Relative $\mathbb{Z}_2^A$ Charge | 0 | 0 |
| | Relative $\mathbb{Z}_4^\Gamma$ Charge | 2 | 0 |
| $\mathbb{Z}_4^\Gamma$-TBC: | GSD | $L = \text{odd} : 2; \quad L = \text{even} : 4$ | 1 |
| | Relative $\mathbb{Z}_2^A$ Charge | 1 | 0 |
| | Relative $\mathbb{Z}_4^\Gamma$ Charge | 1 or 3 | 0 |
| OBC: | GSD | $\geq 2$ | 1 |

Thus the number of ground states of the $\mathbb{Z}_4^\Gamma$ igSPT under periodic boundary condition is also given by (40).

Let us further discuss the $\mathbb{Z}_4$ charge of the ground state. Denote the ground states of (32), (34) and the (38) as $|\text{GS}\rangle_{\text{LG}}$, $|\text{GS}\rangle_{\text{pre}}$ and $|\text{GS}\rangle$ respectively. Suppose the $\mathbb{Z}_2^G$ charge of $|\text{GS}\rangle_{\text{LG}}$ in the Levin-Gu model (32) is $q_{\text{LG}}$, then by definition we have

$$U_{DW} U_\Gamma U_{DW}^\dagger |_{\text{low}} |\text{GS}\rangle_{\text{LG}} = (-1)^{q_{\text{LG}}} |\text{GS}\rangle_{\text{LG}} . \tag{41}$$

As $\mathbb{Z}_2^A$ decouples from the low energy, we also have $U_{DW} U_\Gamma U_{DW}^\dagger |\text{GS}\rangle_{\text{pre}} = (-1)^{q_{\text{LG}}} |\text{GS}\rangle_{\text{pre}}$. Since $|\text{GS}\rangle_{\text{pre}} = U_{DW} |\text{GS}\rangle$, we can then compute the $\mathbb{Z}_4^\Gamma$ charge of $|\text{GS}\rangle$ via,

$$U_\Gamma |\text{GS}\rangle = U_{DW}^\dagger (U_{DW} U_\Gamma U_{DW}^\dagger) |\text{GS}\rangle_{\text{pre}} = (-1)^{q_{\text{LG}}} U_{DW}^\dagger |\text{GS}\rangle_{\text{pre}} = e^{i\frac{\pi}{2} \cdot 2 q_{\text{LG}}} |\text{GS}\rangle . \tag{42}$$

So the $\mathbb{Z}_4^\Gamma$ charge $q$ of the ground state $|\text{GS}\rangle$ is related to the $\mathbb{Z}_2^G$ charge of $|\text{GS}\rangle_{\text{LG}}$ via $q = 2 q_{\text{LG}} \mod 4$.

We are left to determine the symmetry charge of the Levin-Gu model, $q_{\text{LG}}$. While the ground-state degeneracy was obtained exactly in Eq. (40) by the Jordan-Wigner transformation as discussed in Appendix B.1, we could not find $q_{\text{LG}}$ from the Jordan-Wigner transformation. Nevertheless, we can utilize an alternative mapping to the XX chain as discussed in Appendix B.2, to determine $q_{\text{LG}}$ for even $L$'s. The analytical result for even $L$'s was confirmed by exact numerical diagonalization for small $L$'s, which also gives $q_{\text{LG}}$ for odd $L$'s. As a result, extending the $L \mod 4$ dependence of the ground-state degeneracy (40), we find that the symmetry charge of the Levin-Gu model $q_{\text{LG}}$ depends on $\alpha = L \mod 8$: $q_{\text{LG}} = 0$ for $\alpha = 0, 1, 7$, while $q_{\text{LG}} = 1$ for $\alpha = 3, 4, 5$. As presented in Eq. (40), for $\alpha = 2, 6$, the ground states are two fold degenerate. We find that, each of the two degenerate ground states has $q_{\text{LG}} = 0$ and $q_{\text{LG}} = 1$.

We conclude that the $\mathbb{Z}_4^\Gamma$ charge $q$ of ground state of (38) is

$$U_\Gamma |\text{GS}\rangle = e^{i\pi q/2} |\text{GS}\rangle , \qquad q = 2 q_{\text{LG}} = \begin{cases} 0, & \alpha = 0, 1, 7, \\ 2, & \alpha = 3, 4, 5, \\ 0 \,\&\, 2, & \alpha = 2, 6. \end{cases} \tag{43}$$

This is consistent with that the ground state satisfies

$$\tau^x_{i+\frac{1}{2}} |\text{GS}\rangle^{\mathbb{Z}^{\Gamma}_4}_{\text{tw}} = \sigma^z_i \sigma^z_{i+1} |\text{GS}\rangle^{\mathbb{Z}^{\Gamma}_4}_{\text{tw}} \qquad (1 \leq i \leq L). \tag{44}$$

From the above result, it appears that the ground state degeneracy is not well defined in the limit $L \to \infty$. While we do not completely understand the physical mechanism behind the periodic dependence of the ground-state degeneracy on the system size, the ground-state degeneracy for $\alpha = 2, 6$ might be interpreted as a consequence of an effective twist [58]. The effective twist can be seen by mapping the Levin-Gu model to an XX chain. In appendix B, we showed that under a unitary transformation, the Levin-Gu model with PBC can be mapped to an XX chain with PBC and one ground state when $L \in 4\mathbb{Z}$, and XX chain with the twisted boundary condition and two degenerate ground states when $L \in 4\mathbb{Z} + 2$. This is analogous to the phenomenon that an antiferromagnetic chain of odd length is effectively subject to a twisted boundary condition. Here we simply consider the sequence of systems only with $\alpha \in \{0, 1, 3, 4, 5, 7\}$. This would be reasonable if the ground-state degeneracy for $\alpha = 2, 6$ is indeed due to an effective twist; we just consider the sequence of effectively untwisted systems.[16] Then the ground state degeneracy in the thermodynamic limit is regarded as one, consistent with our definition of igSPT.

There still remains the periodic dependence of the $\mathbb{Z}^{\Gamma}_4$ charge in the ground state on the system size: for $\alpha = 0, 1, 7$, the ground state is neutral under $\mathbb{Z}^{\Gamma}_4$, while $\alpha = 3, 4, 5$, the ground state gets a minus sign under the $\mathbb{Z}^{\Gamma}_4$ transformation. However, this minus sign can always be absorbed by suitably modifying the definition of $U_{\Gamma}$ in (39). In fact, in the following sections, we will only be interested in the relative charge of the ground state between the periodic and twisted boundary conditions, which turns out to be system-size independent.

### 3.2.2  Twisted boundary condition

We further discuss the charge of the ground state under the TBC. We can either twist by $\mathbb{Z}^{\Gamma}_4$, or its normal subgroup $\mathbb{Z}^A_2$.

**Twist by $\mathbb{Z}^A_2$:**   We twist the boundary condition by $\mathbb{Z}^A_2$ (labeled by $\mathbb{Z}^A_2$ TBC). The Hamiltonian is

$$\begin{aligned}
H^{\mathbb{Z}^A_2}_{\text{igSPT}} = &-\sum_{i=1}^{L-1} \left( \tau^z_{i-\frac{1}{2}} \sigma^x_i \tau^z_{i+\frac{1}{2}} + \tau^y_{i-\frac{1}{2}} \sigma^x_i \tau^y_{i+\frac{1}{2}} + \sigma^z_i \tau^x_{i+\frac{1}{2}} \sigma^z_{i+1} \right) \\
&+ \tau^z_{L-\frac{1}{2}} \sigma^x_L \tau^z_{\frac{1}{2}} + \tau^y_{L-\frac{1}{2}} \sigma^x_L \tau^y_{\frac{1}{2}} - \sigma^z_L \tau^x_{\frac{1}{2}} \sigma^z_1 \\
= &\ \sigma^z_L H_{\text{igSPT}} \sigma^z_L,
\end{aligned} \tag{45}$$

where $H_{\text{igSPT}}$ is (38). We have already encountered the same algebra below (16). Denote the ground state of $H_{\text{igSPT}}$ and $H^{\mathbb{Z}^A_2}_{\text{igSPT}}$ as $|\text{GS}\rangle$ and $|\text{GS}\rangle^{\mathbb{Z}^A_2}_{\text{tw}}$, respectively. Then we have $|\text{GS}\rangle^{\mathbb{Z}^A_2}_{\text{tw}} = \sigma^z_L |\text{GS}\rangle$. As $U_{\Gamma} \sigma^z_L U^{\dagger}_{\Gamma} = -\sigma^z_L$, we find

$$U_{\Gamma} |\text{GS}\rangle^{\mathbb{Z}^A_2}_{\text{tw}} = -e^{i\pi q/2} |\text{GS}\rangle^{\mathbb{Z}^A_2}_{\text{tw}} = e^{i\pi(q+2)/2} |\text{GS}\rangle^{\mathbb{Z}^A_2}_{\text{tw}}, \tag{46}$$

where $q$ is the $\mathbb{Z}_4$ charge of $|\text{GS}\rangle$ under PBC, given by (43). (46) means that the $\mathbb{Z}^{\Gamma}_4$ charge of the ground state with the $\mathbb{Z}^A_2$ twisted boundary condition differs from that with the periodic boundary condition by two. We thus define the difference between the $\mathbb{Z}^{\Gamma}_4$ charge under $\mathbb{Z}^A_2$-TBC and that under PBC to be the *relative $\mathbb{Z}^{\Gamma}_4$ charge*, which is two. Relative charge is more

---

[16]See also [59] for the system size dependent ground state degeneracy in the $(1 + 1)$d Luttinger liquids.

physical since there are ambiguities in defining the absolute charge as we noticed in the previous subsection. The nontrivial relative $\mathbb{Z}_2^A$ charge shows that the igSPT we constructed in (38) is topologically nontrivial. We also note that the proof applies to all the states.

We remark that the second equality of (45) does not hold under a $\mathbb{Z}_4^\Gamma$ symmetric perturbation. However, the conclusion that the relative $\mathbb{Z}_4^\Gamma$ charge of the low lying states between the $\mathbb{Z}_2^A$ twisted and untwisted sectors being two still holds under a $\mathbb{Z}_4^\Gamma$ symmetric perturbation. We defer this to a subsequent paper [33].

**Twist by $\mathbb{Z}_4^\Gamma$:** We further use the $\mathbb{Z}_4^\Gamma$ symmetry to twist the boundary condition (labeled by $\mathbb{Z}_4^\Gamma$ TBC). The Hamiltonian is

$$H_{\text{igSPT}}^{\mathbb{Z}_4^\Gamma} = -\sum_{i=1}^{L-1}\left(\tau_{i-\frac{1}{2}}^z \sigma_i^x \tau_{i+\frac{1}{2}}^z + \tau_{i-\frac{1}{2}}^y \sigma_i^x \tau_{i+\frac{1}{2}}^y + \sigma_i^z \tau_{i+\frac{1}{2}}^x \sigma_{i+1}^z\right)$$
$$- \tau_{L-\frac{1}{2}}^z \sigma_L^x \tau_{\frac{1}{2}}^y + \tau_{L-\frac{1}{2}}^y \sigma_L^x \tau_{\frac{1}{2}}^z + \sigma_L^z \tau_{\frac{1}{2}}^x \sigma_1^z. \tag{47}$$

The ground state $|\text{GS}\rangle_{\text{tw}}^{\mathbb{Z}_4^\Gamma}$ satisfies

$$\tau_{i+\frac{1}{2}}^x |\text{GS}\rangle_{\text{tw}}^{\mathbb{Z}_4^\Gamma} = \sigma_i^z \sigma_{i+1}^z |\text{GS}\rangle_{\text{tw}}^{\mathbb{Z}_4^\Gamma} \qquad (1 \le i \le L-1), \qquad \tau_{\frac{1}{2}}^x |\text{GS}\rangle_{\text{tw}}^{\mathbb{Z}_4^\Gamma} = -\sigma_L^z \sigma_1^z |\text{GS}\rangle_{\text{tw}}^{\mathbb{Z}_4^\Gamma}. \tag{48}$$

We then measure the $\mathbb{Z}_2^A$ charge using $U_A$ in (36),

$$U_A |\text{GS}\rangle_{\text{tw}}^{\mathbb{Z}_4^\Gamma} = -\prod_{i=1}^{L-1}\left(\sigma_i^z \sigma_{i+1}^z\right)\sigma_L^z \sigma_1^z |\text{GS}\rangle_{\text{tw}}^{\mathbb{Z}_4^\Gamma} = -|\text{GS}\rangle_{\text{tw}}^{\mathbb{Z}_4^\Gamma}, \tag{49}$$

which means that the ground state carries $\mathbb{Z}_2^A$ charge 1. This also implies that if $|\text{GS}\rangle_{\text{tw}}^{\mathbb{Z}_4^\Gamma}$ is an eigenstate of $U_\Gamma$, then it should carry $\mathbb{Z}_4^\Gamma$ charge 1 mod 4 or 3 mod 4.

In fact, by exact numerical diagonalization, we find that there are two degenerate ground states if $L$ is odd and four if $L$ is even. If we organize them into eigenstates of $\mathbb{Z}_4^\Gamma$, half of them have charge 1 mod 4 and the other half have charge 3 mod 4. Since there are different charges, an arbitrary linear combination of them is generically not an $\mathbb{Z}_4^\Gamma$ eigenstate. However, as all of the ground states have $\mathbb{Z}_2^A$ charge 1, an arbitrary linear combination of them also has $\mathbb{Z}_2^A$ charge 1.

From (43), the $\mathbb{Z}_2^A$ charge of the ground state under PBC is always trivial, independent of the system size. Moreover, as we find in (49) the $\mathbb{Z}_2^A$ charge of the ground state under $\mathbb{Z}_4^\Gamma$ TBC is one, independent of the system size. We thus found that the *relative $\mathbb{Z}_2^A$ charge* is size-independent, and it shows that the igSPT we constructed in (38) is topologically nontrivial. Since (44) and (48) also hold for all low energy states with energy $E-E_{\text{GS}} \ll 1$, the above proof of nontrivial $\mathbb{Z}_2^A$ charge of the ground state also applies to low energy states, which should be stable under perturbations.

In summary, we have checked that using either $\mathbb{Z}_2^A$-TBC or $\mathbb{Z}_4^\Gamma$-TBC one can probe the topological nontriviality of the $\mathbb{Z}_4^\Gamma$ igSPT.[17]

---

[17] We need to show that the symmetry features shown in this section is stable under deformation. It was noticed in [15] that if the deformation leads to an accidental symmetry $H$ which has a mixed anomaly with $\mathbb{Z}_4^\Gamma$, then the properties in this section can change due to the mixed anomaly. Assuming such accidental symmetry is the only mechanism that can change the symmetry properties in this section (which is a common lore, and has been discussed recently in [60]), we therefore need to demand that the deformation does not lead to any such accidental symmetry.

### 3.2.3 Open boundary condition

We proceed to discuss the ground state degeneracy under the OBC. When placing the $\mathbb{Z}_4$ igSPT on an open chain, as in section 2.3.3, we let the left most spin be $\sigma$ spin, and right most spin be $\tau$ spin. The Hamiltonian is

$$H_{\text{igSPT}}^{\text{OBC}} = -\sum_{i=2}^{L}\left(\tau_{i-\frac{1}{2}}^z \sigma_i^x \tau_{i+\frac{1}{2}}^z + \tau_{i-\frac{1}{2}}^y \sigma_i^x \tau_{i+\frac{1}{2}}^y\right) - \sum_{i=1}^{L-1}\sigma_i^z \tau_{i+\frac{1}{2}}^x \sigma_{i+1}^z, \tag{50}$$

and the symmetry operator is

$$U_\Gamma = \prod_{i=1}^{L}\sigma_i^x \prod_{i=1}^{L}\exp\left(\frac{i\pi}{4}\left(1 - \tau_{i+\frac{1}{2}}^x\right)\right). \tag{51}$$

We find that the set of operators $\{\sigma_1^z, \sigma_L^z \tau_{L+\frac{1}{2}}^x, U_\Gamma\}$ commute with the Hamiltonian (50). The irreducible representation of the above algebra is two, hence the ground states of (50) are at least two fold degenerate. In appendix D.2, we show that the ground state degeneracy is four for $L \in 2\mathbb{Z} + 1$, and two for $L \in 2\mathbb{Z}$.

## 3.3 Stability of igSPT

As discussed in section 2, the $\mathbb{Z}_2^A \times \mathbb{Z}_2^G$ gSPT is unstable upon perturbation towards the gapped SPT phase. It immediately enters the $\mathbb{Z}_2^A \times \mathbb{Z}_2^G$ gapped SPT phase when transverse field $\lambda$ passes the critical value $\lambda_c = 1$. How about the stability of the $\mathbb{Z}_4^\Gamma$ igSPT against perturbation into a gapped phase with a unique ground state?

First of all, since $\mathbb{Z}_4^\Gamma$ is non-anomalous, in principle, there is no obstruction to deform the system to $\mathbb{Z}_4$ symmetric gapped phase with a unique ground state [61]. Secondly, since there is no $\mathbb{Z}_4^\Gamma$ gapped SPT, the only gapped phase with a non-degenerate ground state is the trivially gapped phase. In this subsection, we will examine the most obvious $\mathbb{Z}_4^\Gamma$ symmetric perturbation that can drive the igSPT into a trivially gapped phase,

$$-h\sum_{i=1}^{L}\left(\sigma_i^x + \tau_{i+\frac{1}{2}}^x\right), \tag{52}$$

where $h > 0$. When $h \gg 1$, as $\tau_{i+\frac{1}{2}}^x$ anticommutes with the first and second term of the Hamiltonian (38), and $\sigma_i^x$ anticommutes with the third term, only (52) survives and it is in the trivially gapped phase. This means that there must be at least one phase transition as $h$ increases from zero where either the $\mathbb{Z}_4^\Gamma$ charge under PBC, or the relative $\mathbb{Z}_2^A$ charge under $\mathbb{Z}_4^\Gamma$-TBC or relative $\mathbb{Z}_4^\Gamma$ charge under $\mathbb{Z}_2^A$-TBC jumps.

After adding (52), the proof of non-trivial (relative) charges under the twisted boundary conditions in Section 3.2.2 no longer apply. We therefore numerically calculate the charges as a function of $h$, and indeed observe the jumps for finite $h$. We relegate the details of small-scale numerical study in Appendix F. In a subsequent work [33], by using the Kennedy-Tasaki transformation, we analytically show that the phase transition happens at finite $h$, therefore shows that igSPT (38) is stable under the perturbation (52).

# 4 gSPT and IgSPT in the spin-1 system

In this section, we briefly introduce a more realistic spin-1 model which hosts the igSPT and gSPT simultaneously. This model is studied in detail in [30] by one of the authors in this work (L.L.) together with Yang, Okunishi and Katsura. We briefly review the results there, and fit them into our framework.

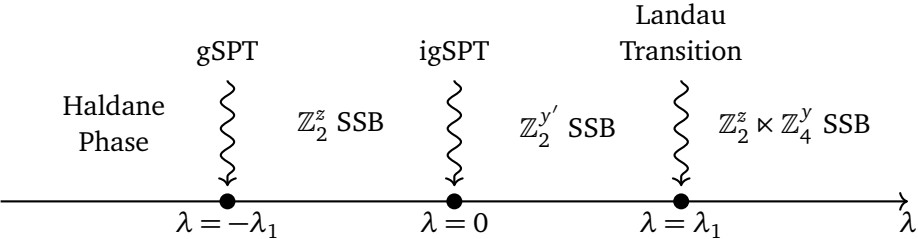

Figure 4: The phase diagram of (53) when $\theta=0$.

## 4.1 The model and phase diagram

The Hamiltonian is given by

$$H(\theta, \lambda) = (1 - \lambda)H_{\text{BLBQ}} + (1 + \lambda)U_{KT}H_{\text{BLBQ}}U_{KT}^{\dagger}, \quad -\frac{\pi}{4} < \theta < \arctan\frac{1}{2}, \tag{53}$$

where

$$H_{\text{BLBQ}} = \cos\theta(\vec{S}_i \cdot \vec{S}_{i+1}) + \sin\theta(\vec{S}_i \cdot \vec{S}_{i+1})^2, \tag{54}$$

$$U_{KT} = \prod_{\mu < \nu} \exp(i\pi S_{\mu}^z S_{\nu}^x). \tag{55}$$

$\vec{S}$ is spin-1 operator. $U_{KT}$ is a non-local unitary operator implementing the Kennedy-Tasaki (KT) transformation [48–50]. Under the KT transformation, $\lambda \leftrightarrow -\lambda$, and $\lambda = 0$ is the self-dual point. For each $\theta$ and $\lambda$, the Hamiltonian (53) preserves three global symmetries:

1. $\mathbb{Z}_2^z$: $\pi$ rotation in $z$ direction, generated by $\prod_j e^{i\pi S_j^z}$.

2. $\mathbb{Z}_4^y$: $\pi/2$ rotation in $y$ direction, generated by $\prod_j e^{i\frac{\pi}{2}S_j^y}$.

3. $\mathbb{Z}^{\mathbf{T}}$: translation symmetry.

The phase diagram of $\theta = 0$ is obtained in [30], as shown in figure 4. See [30] for the full 2d phase diagram in the $(\lambda, \theta)$ plane.

## 4.2 $\mathbb{Z}_2^z \ltimes \mathbb{Z}_4^y \times \mathbb{Z}^{\mathbf{T}}$ igSPT

Let us start by discussing the self-dual point $\lambda = 0$ which we argue to be a igSPT. Taking the low energy limit around this point, some degrees of freedom decouple, and the 3-dimensional Hilbert space per site in the spin-1 model reduces to 2-dimensional Hilbert space per site, hence effectively becomes a spin-$\frac{1}{2}$ model. The spin-$\frac{1}{2}$ Hamiltonian turns out to be the XXZ model [30]:

$$H(\lambda \ll 1) = -(1 + \lambda)\sum_{j=1}^L \sigma_j^x \sigma_{j+1}^x + (1 - \lambda)\sum_{j=1}^L \sigma_j^y \sigma_{j+1}^y. \tag{56}$$

This model also has three global symmetries:

1. $\mathbb{Z}_2^{z'}$: generated by $\prod_i \sigma_i^z$.

2. $\mathbb{Z}_2^{y'}$: generated by $\prod_i i\sigma_i^y$.

3. $\mathbb{Z}^{\mathbf{T}}$: translation symmetry.

We use the primes to distinguish the symmetries of the spin-1/2 model from those of the spin-1 model. Denote their background fields as $A'_z, A'_y$ and $A_T$. The symmetries $\mathbb{Z}_2^{z'}, \mathbb{Z}_2^{y'}$ and $\mathbb{Z}_2^{\mathbf{T}} \subset \mathbb{Z}^{\mathbf{T}}$ have a mixed anomaly [58, 62, 63] whose inflow action is

$$\omega_{3d} = e^{i\pi \int_{M_3} A'_y A'_z A_T} . \tag{57}$$

However, in the entire Hibert space of spin-1 system, the $\mathbb{Z}_2^{z'} \times \mathbb{Z}_2^{y'}$ is realized as $\mathbb{Z}_2^z \ltimes \mathbb{Z}_4^y$ symmetry with the following extension:

$$Y'Z' = R_\pi^y Z'Y', \tag{58}$$

where $R_\pi^y = \prod_{j=1}^L \exp(i\pi S_j^y)$, $Y' = \prod_{j=1}^L \exp(i\pi S_j^y/2)$ and $Z' = \prod_{j=1}^L \exp(i\pi S_j^z)$. $\exp(i\pi S_j^y)$ has eigenvalues $\{-1, -1, 1\}$. In the low energy limit, the spin-$\frac{1}{2}$ model only acts nontrivially on the first two components of the spin-1 Hilbert space under the eigenbasis of $\exp(i\pi S_j^y)$, hence $\exp(i\pi S_j^y) = -1$ in the spin-$\frac{1}{2}$ model, $Y', Z'$ in (58) reduces to the standard spin-$\frac{1}{2}$ operators $\sigma_j^z = \exp(i\pi S_j^z)$ and $i\sigma_j^y = \exp(i\pi S_j^y/2)$. In terms of the background fields, (58) gives us the restriction

$$dA_Y = A'_y A'_z \mod 2, \tag{59}$$

where $A_Y$ is 1-cochain for $\mathbb{Z}_2^Y$ normal subgroup of $\mathbb{Z}_4^y$ symmetry. In summary, we can identify $\mathbb{Z}_2^{z'}$ and $\mathbb{Z}_2^{y'}$ in the spin-$\frac{1}{2}$ theory with the $\mathbb{Z}_2^z$ and $\mathbb{Z}_4^y/\mathbb{Z}_2^Y$ in the spin-1 theory respectively.

Besides, since $\exp(i\pi S_j^y) = -1$ for each site in the low energy sector, the ground state is stacked by a weak gapped SPT phase protected by translation and $\mathbb{Z}_2^y$ symmetry [64]. This is represented by the topological action $e^{i\pi \int_{M_2} A_Y A_T}$ and by (59), it depends on the extension to a 3d bulk $M_3$,

$$e^{i\pi \int_{M_2} A_Y A_T} = e^{i\pi \int_{M_3} A'_y A'_z A_T} . \tag{60}$$

This induced anomaly from stacking a weak gapped SPT phase cancels against the mixed anomaly (57) in the low energy. Thus the total spin-1 system is anomaly free. This shows that the spin-1 system is a igSPT, protected by the total symmetry $\mathbb{Z}_2^z \ltimes \mathbb{Z}_4^y \times \mathbb{Z}^{\mathbf{T}}$.

The total symmetry can be decomposed into two extensions,

$$1 \to \mathbb{Z}_2^z \times \mathbb{Z}_2^Y \times \mathbb{Z}_2^{\mathbf{T}} \to \mathbb{Z}_2^z \ltimes \mathbb{Z}_4^y \times \mathbb{Z}_2^{\mathbf{T}} \to \mathbb{Z}_2^{y'} \to 1, \tag{61}$$

and

$$1 \to \mathbb{Z}_4^y \to \mathbb{Z}_2^z \ltimes \mathbb{Z}_4^y \times \mathbb{Z}_2^{\mathbf{T}} \to \mathbb{Z}_2^z \times \mathbb{Z}_2^{\mathbf{T}} \to 1. \tag{62}$$

Note that (62) is still a nontrivial extension.[18] Comparing with (1), we see that the $\mathbb{Z}_2^z \ltimes \mathbb{Z}_4^y \times \mathbb{Z}^{\mathbf{T}}$ igSPT can be constructed either by starting with $G = \mathbb{Z}_2^{y'}$ SSB phase or $G = \mathbb{Z}_2^z$ SSB phase, which exactly correspond to the regimes $\lambda > 0$ and $\lambda < 0$ in figure 4. Moreover, from (57), the anomalous symmetries in the low energy are $\widehat{\Gamma} = \mathbb{Z}_2^{y'} \times \mathbb{Z}_2^{z'} \times \mathbb{Z}_2^{\mathbf{T}}$. This provides an example where the SSB symmetry $G$ is strictly smaller than the anomalous symmetry $\widehat{\Gamma}$, which generalizes the construction in [16].

---

[18]Both (61) and (62) are not central extensions, since both $\mathbb{Z}_2^z \times \mathbb{Z}_2^Y$ and $\mathbb{Z}_4^y$ are not center subgroup of $\mathbb{Z}_2^z \ltimes \mathbb{Z}_4^y$.

### 4.3 $\mathbb{Z}_2^z \ltimes \mathbb{Z}_4^y \times \mathbb{Z}^{\mathbf{T}}$ gSPT

Let us further consider the critical point at $\lambda = -\lambda_1$. The two phases around this critical point are $\mathbb{Z}_2^z$ SSB phase and a nontrivial gapped SPT protected by $\mathbb{Z}_2^z \times \mathbb{Z}_2^Y$, a.k.a. the Haldane phase. This fits into the phase diagram of gSPT in the left panel of figure 1.

Moreover, at $\lambda = -\lambda_1$, the Hamiltonian (53) has a unique ground state under periodic boundary condition for a finite system size but has two ground states under the open boundary condition (up to exponential splitting). There are also three string order parameters with nonzero expectation value in the Haldane phase $O_\mu = \langle S_m^\mu \prod_{m<j<n} \exp(i\pi S_j^\mu) S_n^\mu \rangle$ ($\mu = x, y, z$). When the system is turned into this critical point, only $O_y$ remains nonzero but the other two decay to zero algebraically quickly. All these evidence suggest that the critical point at $\lambda = -\lambda_1$ is a nontrivial gSPT. As the system has total symmetry $\mathbb{Z}_2^z \ltimes \mathbb{Z}_4^y \times \mathbb{Z}^{\mathbf{T}}$, we name the critical point as $\mathbb{Z}_2^z \ltimes \mathbb{Z}_4^y \times \mathbb{Z}^{\mathbf{T}}$ gSPT, although only a subgroup $\mathbb{Z}_2^z \times \mathbb{Z}_2^Y$ protects the gapped SPT in the nearby phase.

## Acknowledgments

L.L. sincerely thanks Jian-da Wu's group for helpful discussions on the Jordan-Winger transformation of the Levin-Gu model; Hong Yang for discussions on the gSPT and igSPT in the Spin-1 system; Yutan Zhang for kind help on the Julia code. Y.Z. sincerely thanks Jie Wang for discussions on stability of gapless systems, Qing-Rui Wang for discussions on decorated defect constructions, and Nick Jones, Ryan Thorngren and Ruben Verresen for helpful comments and discussions. We also thank Atsushi Ueda, Yuan Yao, Kantaro Ohmori for useful discussions.

**Funding information** Y.Z. is partially supported by WPI Initiative, MEXT, Japan at IPMU, the University of Tokyo. This work was supported in part by MEXT/JSPS KAKENHI Grants No. JP17H06462 and No. JP19H01808, and by JST CREST Grant No. JPMJCR19T2.

## A Stability of boundary degeneracy of $\mathbb{Z}_2^A \times \mathbb{Z}_2^G$ gapped SPT

We find in section 2.3.3 that if we suitably change OBC by adding boundary interactions, the ground state degeneracy can be lifted from four to two. In this appendix, we would like to argue that exactly degenerate ground states of the $\mathbb{Z}_2^A \times \mathbb{Z}_2^G$ gapped SPT, which is always four, does not lift under arbitrary symmetric perturbations localized at the boundary.

Let us truncate the system in the same way as section 2.3.3. The $\sigma$ spins are supported on $i = 1, \dots, L$, and the $\tau$ spins are supported on $i + \frac{1}{2} = \frac{3}{2}, \dots, L + \frac{1}{2}$. Let us begin by choosing one particular OBC such that the Hamiltonian is

$$H_{\mathrm{SPT}}^{\mathrm{OBC}} = -\sum_{i=1}^{L-1} \sigma_i^z \tau_{i+\frac{1}{2}}^x \sigma_{i+1}^z - \sum_{i=2}^{L} \tau_{i-\frac{1}{2}}^z \sigma_i^x \tau_{i+\frac{1}{2}}^z . \tag{A.1}$$

Suppose the boundary perturbation at the left end is supported on 2 sites, $1, \frac{3}{2}$. A generic symmetric perturbation takes the form

$$\Delta H_{\mathrm{SPT}}^{\mathrm{OBC}} = (\sigma_1^x)^{\beta_1} (\tau_{\frac{3}{2}}^x)^{\beta_{\frac{3}{2}}} , \tag{A.2}$$

where $\beta_{1,\frac{3}{2}} \in \{0, 1\}$.[19] Let us find the local operators that commute with both $H_{\mathrm{SPT}}^{\mathrm{OBC}}$ and

---

[19] For perturbations supported on 3 sites, one also allows $\sigma_1^z \sigma_2^z$. But for 2 site perturbation, Pauli $Z$ operators are forbidden by the symmetries.

$\Delta H_{\text{SPT}}^{\text{OBC}}$. Any interaction commuting with $H_{\text{SPT}}^{\text{OBC}}$ are composed of the building blocks $\sigma_1^z$, $\sigma_1^x \tau_{\frac{3}{2}}^z$, $\tau_{L+\frac{1}{2}}^x \sigma_L^z$, $\tau_{L+\frac{1}{2}}^z$ and all the terms that already exist in (A.1). Using these building blocks, a generic term that might anticommute with the boundary perturbation takes the form

$$\mathcal{O}^{u_1 u_2 u_3 u_4} = (\sigma_1^z)^{u_1} (\sigma_1^x \tau_{\frac{3}{2}}^z)^{u_2} (\sigma_1^z \tau_{\frac{3}{2}}^x \sigma_2^z)^{u_3} (\tau_{\frac{3}{2}}^z \sigma_2^x \tau_{i+\frac{5}{2}}^z)^{u_4}, \tag{A.3}$$

where $u_{1,2,3,4} \in \{0, 1\}$. Requiring $[\mathcal{O}, \Delta H_{\text{SPT}}^{\text{OBC}}] = 0$, we find that the coefficients need to satisfy the linear equations

$$\beta_1 (u_1 + u_3) + \beta_{\frac{3}{2}} (u_2 + u_4) = 0 \mod 2. \tag{A.4}$$

Note that $\beta_{1,\frac{3}{2}}$ are given, while $u$'s are variables to be determined. There are 4 variables, and one equation, hence one is free to choose arbitrary value of $u_1, u_2$, such that $u_i$'s for $i = 3, 4$ are constrained by the equation. One solution would be $u_3 = \beta_{\frac{3}{2}} - u_1, u_4 = \beta_1 - u_2$. On the other hand, the algebra between the operators $\{\mathcal{O}^{u_1 u_2 u_3 u_4}, U_A, U_B\}$ are

$$\begin{aligned}
\mathcal{O}^{u_1 u_2 u_3 u_4} \mathcal{O}^{u_1' u_2' u_3' u_4'} &= (-1)^{u_1 u_2' + u_1' u_2} \mathcal{O}^{u_1' u_2' u_3' u_4'} \mathcal{O}^{u_1 u_2 u_3 u_4}, \\
U_A \mathcal{O}^{u_1 u_2 u_3 u_4} &= (-1)^{u_2} \mathcal{O}^{u_1 u_2 u_3 u_4} U_A, \\
U_G \mathcal{O}^{u_1 u_2 u_3 u_4} &= (-1)^{u_1} \mathcal{O}^{u_1 u_2 u_3 u_4} U_G.
\end{aligned} \tag{A.5}$$

The commutation relations only depends on $u_1, u_2$! Hence we are free to choose two commuting independent operators $\mathcal{O}^{10 u_3 u_4}$ and $\mathcal{O}^{01 u_3' u_4'}$ whose common eigenvalues $(a, b)$ label the ground states $|(a, b)\rangle$, where $u_{3,4}$ and $u_{3,4}'$ are arbitrary solutions of (A.4). The four orthogonal ground states are thus given by

$$|(a, b)\rangle, \quad |(-a, b)\rangle = U_G |(a, b)\rangle, \quad |(a, -b)\rangle = U_A |(a, b)\rangle, \quad |(-a, -b)\rangle = U_A U_G |(a, b)\rangle. \tag{A.6}$$

The above discussion can easily be generalized to perturbation supported on arbitrary number sites. We thus conclude that, for the $\mathbb{Z}_2^A \times \mathbb{Z}_2^G$ gapped SPT, the exact four fold ground state degeneracy on an open chain is stable under boundary perturbation.

# B  Spectrum of Levin-Gu model under different boundary conditions

In this appendix, we show the energy spectrum of Levin-Gu model [42] under different boundary conditions analytically. The analytic results are confirmed by the numerical calculation.

## B.1  Exact solutions under PBC by Jordan-Wigner transformation

The Hamiltonian of Levin-Gu model is

$$H_{\text{LG}} = -\sum_{i=1}^{L} \left( \sigma_i^x - \sigma_{i-1}^z \sigma_i^x \sigma_{i+1}^z \right), \tag{B.1}$$

which respects the $\mathbb{Z}_2$ symmetry generated by

$$U_G = \prod_{i=1}^{L} \sigma_i^x \prod_{i=1}^{L} \exp\left( \frac{i\pi}{4} (1 - \sigma_i^z \sigma_{i+1}^z) \right). \tag{B.2}$$

We apply the Jordan-Wigner (JW) transformation which maps spin operator to fermion operator

$$\sigma_i^x = (-1)^{n_i} = 1 - 2 f_i^\dagger f_i, \qquad \sigma_i^z = \prod_{j=1}^{i-1} (-1)^{n_j} (f_i^\dagger + f_i), \tag{B.3}$$

where $n_i := f_i^\dagger f_i$ is fermion density operator. Note that when $i = 1$, we simply have $\sigma_1^z = f_1^\dagger + f_1$. We also assume PBC of the spins, i.e. $\sigma_i^a = \sigma_{i+L}^a$.

Applying the JW transformation to the Levin-Gu model, we can rewrite (B.1) in terms of the fermions,

$$H_{\text{LG}} = -L + \sum_{i=1}^{L} \left( 2f_i^\dagger f_i + (f_i^\dagger - f_i)(f_{i+2}^\dagger + f_{i+2}) \right), \tag{B.4}$$

with boundary condition

$$f_{i+L} = -(-1)^F f_i, \qquad F = \sum_{j=1}^{L} n_j. \tag{B.5}$$

After Fourier transformation and Bogoliubov transformation, this Hamiltonian is diagonal

$$H_{\text{LG}} = \sum_k \omega_k \left( c_k^\dagger c_k - \frac{1}{2} \right), \qquad (-1)^{\sum_k c_k^\dagger c_k} = (-1)^F, \tag{B.6}$$

where $\omega_k = 4|\cos k|$. There are zero modes if $k$ can be either $\frac{\pi}{2}$ or $\frac{3\pi}{2}$, and whether they are realizable depends on the boundary condition. It turns out that depending on $L \in 4\mathbb{Z}, 4\mathbb{Z} + 2$ or $2\mathbb{Z} + 1$, the boundary condition behaves differently. We discuss them separately.

**Case 1: $L \in 4\mathbb{Z}$**

If $(-1)^F = -1$, the fermion chain has PBC. This means $k = \frac{2\pi j}{L}$ where $j = 0, \cdots, L-1$. Therefore, when $j = \frac{L}{4}$ and $j = \frac{3L}{4}$, we have two zero modes at $k = \frac{\pi}{2}$ and $k = \frac{3\pi}{2}$. Since $(-1)^F = -1$, the ground states are: $c_{\frac{\pi}{2}}^\dagger |\text{VAC}\rangle_{\text{PBC}}$ and $c_{\frac{3\pi}{2}}^\dagger |\text{VAC}\rangle_{\text{PBC}}$. The ground state energy is

$$E_{\text{GS}}^{\text{PBC}} = -2 \sum_{j=0}^{L-1} \left| \cos\left( \frac{2\pi j}{L} \right) \right| = -4 \cot\left( \frac{\pi}{L} \right). \tag{B.7}$$

If $(-1)^F = 1$, the fermion chain has anti-periodic boundary condition (ABC) where $k = \frac{(2j+1)\pi}{L}$. Since $L \in 4\mathbb{Z}$, there is no zero mode. the ground state is $|\text{VAC}\rangle_{\text{ABC}}$ with ground state energy:

$$E_{\text{GS}}^{\text{ABC}} = -2 \sum_{j=0}^{L-1} \left| \cos\left( \frac{(2j+1)\pi}{L} \right) \right| = -\frac{4}{\sin\left( \frac{\pi}{L} \right)}. \tag{B.8}$$

As $E_{\text{GS}}^{\text{ABC}} < E_{\text{GS}}^{\text{PBC}}$, the Levin-Gu model has an unique true ground state which is vacuum of ABC after Jordan-Wigner transformation.

**Case 2: $L \in 4\mathbb{Z} + 2$**

If $(-1)^F = -1$, the fermion chain has PBC where $k = \frac{2\pi j}{L}$, $j = 0, \cdots, L-1$. Since $L = 4m + 2 \in 4\mathbb{Z} + 2$, there is no zero mode. The ground states are $c_{\frac{2m\pi}{4m+2}}^\dagger |\text{VAC}\rangle_{\text{PBC}}$, $c_{\frac{2\pi(m+1)}{4m+2}}^\dagger |\text{VAC}\rangle_{\text{PBC}}$, $c_{\frac{2\pi(3m+1)}{4m+2}}^\dagger |\text{VAC}\rangle_{\text{PBC}}$ and $c_{\frac{2\pi(3m+2)}{4m+2}}^\dagger |\text{VAC}\rangle_{\text{PBC}}$. The ground state energy is

$$E_{\text{GS}}^{\text{PBC}} = -2 \sum_{j=0}^{L-1} \left| \cos\left( \frac{2\pi j}{L} \right) \right| + 4\cos(\frac{m\pi}{2m+1}) = -\frac{4}{\sin\left( \frac{\pi}{L} \right)} + 4\sin\left( \frac{\pi}{L} \right). \tag{B.9}$$

If $(-1)^F = 1$, the fermion chain has ABC where $k = \frac{(2j+1)\pi}{L}$. Since $L = 4m + 2 \in 4\mathbb{Z} + 2$, there are two zero modes at $j = m$ and $j = 3m + 1$. The ground states are double degenerate

$|\text{VAC}\rangle_{\text{ABC}}$ and $c^\dagger_{\frac{\pi}{2}} c^\dagger_{\frac{3\pi}{2}} |\text{VAC}\rangle_{\text{ABC}}$ with energy

$$E_{\text{GS}}^{\text{ABC}} = -2 \sum_{j=0}^{L-1} \left| \cos\left( \frac{(2j+1)\pi}{L} \right) \right| = -4 \cot\left( \frac{\pi}{L} \right). \tag{B.10}$$

Since

$$E_{\text{GS}}^{\text{ABC}} - E_{\text{GS}}^{\text{PBC}} = -4 \cot\left( \frac{\pi}{L} \right) + \frac{4}{\sin\left( \frac{\pi}{L} \right)} - 4 \sin\left( \frac{\pi}{L} \right) = -4 \cot\left( \frac{\pi}{L} \right) \left( 1 - \cos\left( \frac{\pi}{L} \right) \right) < 0, \tag{B.11}$$

the Levin-Gu model has double degenerate ground states which is vacuum of ABC.

**Case 3: $L \in 2\mathbb{Z} + 1$**

If $(-1)^F = 1$, the fermion chains has ABC where $k = \frac{(2j+1)\pi}{L}$ and where $j = 0, \cdots, L-1$. Now since $L = 2m+1 \in 2\mathbb{Z}+1$, there is no zero mode. The ground states is $|\text{VAC}\rangle_{\text{ABC}}$ with energy

$$E_{\text{GS}}^{\text{ABC}} = -2 \sum_{j=0}^{2m} \left| \cos\left( \frac{(2j+1)\pi}{2m+1} \right) \right| = -4 \sum_{j=0}^{m-1} \left| \cos\left( \frac{(2j+1)\pi}{2m+1} \right) \right| - 2. \tag{B.12}$$

If $(-1)^F = -1$, the fermion chain has PBC where $k = \frac{2\pi j}{L}$ where $j = 0, \cdots, L$. Now since $L = 2m+1 \in 2\mathbb{Z}+1$, there is also no zero mode. Here we note that the energy of $|\text{VAC}\rangle_{\text{PBC}}$ is the same as (B.12)

$$E_{\text{VAC}}^{\text{PBC}} = -2 \sum_{j=0}^{2m} \left| \cos\left( \frac{2\pi j}{2m+1} \right) \right| = -4 \sum_{j=1}^{m} \left| \cos\left( \frac{2\pi j}{2m+1} \right) \right| - 2$$

$$= -4 \sum_{j=1}^{m} \left| \cos\left( \frac{(2m-2j+1)\pi}{2m+1} \right) \right| - 2 = -4 \sum_{j=0}^{m-1} \left| \cos\left( \frac{(2j+1)\pi}{2m+1} \right) \right| - 2. \tag{B.13}$$

Since there is no zero mode, the ground state energy in $(-1)^F = -1$ sector must be higher than $E_{\text{VAC}}^{\text{PBC}}$ which coincides with the ground state energy (B.12) under the ABC and the unique true ground state is $|\text{VAC}\rangle_{\text{ABC}}$.

In summary, the ground state degeneracy of the Levin-Gu model under PBC is two if $L \in 4\mathbb{Z}+2$, and one otherwise. This proves (40).

## B.2 Mapping to XX chain and charge of ground state

When the system size is even ($L = 2m$), there is a unitary transformation [7]

$$U = \prod_{j=1}^{m} \exp\left( \frac{\pi i}{2} \sigma_{2j}^y \right) \prod_{j=1}^{m} i \frac{\sigma_{2j}^z + \sigma_{2j}^x}{\sqrt{2}} \prod_{j=1}^{m} \exp\left( \frac{\pi i \left(1 - \sigma_{2j-1}^z\right)\left(1 - \sigma_{2j}^z\right)}{4} \right), \tag{B.14}$$

which maps the Levin-Gu model to a XX chain with imaginary hopping constant.

$$UH_{\text{LG}}U^\dagger = -\sum_{j=1}^{m} (\sigma_{2j-1}^z \sigma_{2j}^x - \sigma_{2j}^x \sigma_{2j+1}^x - \sigma_{2j-1}^x \sigma_{2j}^z + \sigma_{2j}^z \sigma_{2j+1}^x)$$

$$= -\sum_{j=1}^{L} i \sigma_j^+ \sigma_{j+1}^- + h.c., \tag{B.15}$$

where $\sigma_j^+ = \sigma_j^z + i\sigma_j^x$. The imaginary hopping XX chain can be further mapped to a standard XX chain by a unitary transformation

$$U_1 = \prod_{j=1}^{L} \exp\left(\frac{\pi i}{2} j \sigma_j^y\right). \tag{B.16}$$

The resulting Hamiltonian is

$$U_1 U H_{\text{LG}} U^\dagger U_1^\dagger = -\sum_{j=1}^{L} (\sigma_j^z \sigma_{j+1}^z + \sigma_j^x \sigma_{j+1}^x), \tag{B.17}$$

with boundary condition

$$\sigma_{L+j}^z = i^L \sigma_j^z, \qquad \sigma_{L+j}^x = i^L \sigma_j^x. \tag{B.18}$$

After taking the continuum limit [65, 66]

$$(\sigma^z + i\sigma^x) \propto e^{i\theta}, \qquad \sigma^y \propto \frac{a}{2\pi} \partial_x \phi, \tag{B.19}$$

the low energy theory of standard XX chain is the free boson theory and the energy of eigenstate $|m, n\rangle$ is[20]

$$(E_{m,n} - E_{0,0}) \propto \frac{\pi}{2L}(4m^2 + n^2), \tag{B.20}$$

where the integer pairs $(m, n)$ are determined by the boundary conditions $\theta(x+L) = \theta(x) + 2\pi m$ and $\phi(x+L) = \phi(x) + 2\pi n$. By combining (B.17), (B.18) and (B.20), we conclude as follows.

1. When $L \in 4\mathbb{Z}$, the Levin-Gu model is equivalent to the XX chain with PBC where $m \in \mathbb{Z}$ and $n \in \mathbb{Z}$. Its energy minimizes at a unique value $(m, n) = (0, 0)$, and the unique ground state is $|0, 0\rangle$.

2. When $L \in 4\mathbb{Z} + 2$, the Levin-Gu model is equivalent to the XX chain with ABC where $m \in \mathbb{Z} + 1/2$ and $n \in \mathbb{Z}$. Its energy minimizes at two distinct values $(m, n) = (\pm\frac{1}{2}, 0)$, and there are two degenerate ground states $|\pm\frac{1}{2}, 0\rangle$.

This is consistent with the results from JW transformation in (B.1).

Moreover we can obtain the $\mathbb{Z}_2$ symmetry (B.2) after transformation

$$U_G' = U_1 U U_G U^\dagger U_1^\dagger = \prod_{j=1}^{L} \sigma_j^y \prod_{j=1}^{\frac{L}{2}} \exp\left(\frac{\pi i}{4}(2 + \sigma_{2j-1}^x \sigma_{2j}^z - \sigma_{2j}^z \sigma_{2j+1}^x)\right). \tag{B.21}$$

After taking the continuum limit (B.19), the $\mathbb{Z}_2$ symmetry operator in the low energy is given by

$$U_G' = i^{\frac{L}{2}} \exp\left(\frac{i}{2}\int \partial_x \phi \, dx - \frac{i}{2}\int \partial_x \theta \, dx\right). \tag{B.22}$$

The charge of the state can be found by acting $U_G'$ on $|m, n\rangle$,

$$U_G' |m, n\rangle = i^{\frac{L}{2}} e^{i\pi(n-m)} |m, n\rangle. \tag{B.23}$$

Therefore when $L \in 4\mathbb{Z}$, the charge of ground state $|0, 0\rangle$ is $(-1)^{L/4}$. When $L \in 4\mathbb{Z} + 2$ the charges of ground states $|\pm\frac{1}{2}, 0\rangle$ are $\pm(-1)^{\frac{L-2}{4}}$. This proves (42) for even $L$.

---

[20]Since we are only interested in ground state degeneracy, we don't consider excitations of the oscillator modes.

### B.3 Spectrum under open boundary condition

In this section, we use the transformations (B.14) and (B.16) to discuss spectrum of Levin-Gu model under OBC

$$H_{\text{LG}}^{\text{OBC}} = -\sum_{i=2}^{L-1} \left( \sigma_i^x - \sigma_{i-1}^z \sigma_i^x \sigma_{i+1}^z \right). \tag{B.24}$$

There are two boundary operators $\sigma_1^z$ and $\sigma_L^z$ commuting with Hamiltonian.

When $L \in 2\mathbb{Z}$, the Hamiltonian (B.24) and the boundary operators $\sigma_{1,L}^z$ after the transformation are given by

$$U_1 U H_{\text{LG}}^{OBC} U^\dagger U_1^\dagger = -\sum_{j=1}^{\frac{L}{2}-1} \left( \sigma_{2j-1}^x \sigma_{2j}^x + \sigma_{2j+1}^z \sigma_{2j+2}^z + \sigma_{2j}^x \sigma_{2j+1}^x + \sigma_{2j}^z \sigma_{2j+1}^z \right), \tag{B.25}$$

$$U_1 U \sigma_1^z U^\dagger U_1^\dagger = -\sigma_1^x, \qquad U_1 U \sigma_L^z U^\dagger U_1^\dagger = (-1)^{\frac{L}{2}+1} \sigma_L^z. \tag{B.26}$$

After taking the continuum limit, the boundary operators are $-\sin\theta(x = 0)$ and $(-1)^{\frac{L}{2}+1} \cos\theta(x = L)$. As the ground state should be the eigenstate of the boundary operators $-\sigma_1^x, (-1)^{\frac{L}{2}+1} \sigma_L^z, -\sin\theta(x = 0) = \pm 1, (-1)^{\frac{L}{2}+1} \cos\theta(x = L) = \pm 1$. They determine the boundary conditions $\theta(x = 0) = \pm\frac{\pi}{2}$ and $\theta(x = L) = 0$ or $\pi$. The ground state energy under these four boundary conditions are exactly the same.

When $L \in 2\mathbb{Z} + 1$, we only do the transformation (B.14) for even number of sites, say, $i = 1, \dots, L - 1$. We still do $\pi/2$ rotation along $y$ direction, i.e. $U_1$ in (B.16), on the $L$-th site. The Hamiltonian (B.24) and the boundary operators after the transformation are given by

$$U_1 U H_{\text{LG}}^{OBC} U^\dagger U_1^\dagger = -\sum_{j=1}^{\frac{L-1}{2}} \left( \sigma_{2j-1}^x \sigma_{2j}^x + \sigma_{2j}^x \sigma_{2j+1}^x \right) + \sum_{j=1}^{\frac{L-3}{2}} \left( \sigma_{2j+1}^z \sigma_{2j+2}^z + \sigma_{2j}^z \sigma_{2j+1}^z \right), \tag{B.27}$$

$$U_1 U \sigma_1^z U^\dagger U_1^\dagger = -\sigma_1^x, \qquad U_1 U \sigma_L^z U^\dagger U_1^\dagger = \sigma_L^x. \tag{B.28}$$

After taking the continuum limit, the boundary operators are $-\sin\theta(x = 0)$ and $\sin\theta(x = L)$ which implies boundary conditions are $\theta(x = 0) = \pm\frac{\pi}{2}$ and $\theta(x = L) = \pm\frac{\pi}{2}$, and the signs are uncorrelated. Unlike even size, the states with different boundary conditions have different energies,

$$E_{(\mp\frac{\pi}{2}, \pm\frac{\pi}{2})} - E_{(\pm\frac{\pi}{2}, \pm\frac{\pi}{2})} \propto \frac{1}{L}, \tag{B.29}$$

where the signs are correlated. Therefore the true ground states are double degenerate and are in the sector with boundary conditions $\theta(x = 0) = \theta(x = L) = \pm\frac{\pi}{2}$.

## C Equivalence between ground sector of $\mathbb{Z}_4$ igSPT and Levin-Gu model

In this section, we show the ground state of the pre-decorated model (34) of $\mathbb{Z}_4^\Gamma$ igSPT is the same as the Levin-Gu model (32) with $\tau_i^x = 1$.

Let us begin with the pre-decorated model (34) with PBC, which we reproduce here

$$U_{DW} H_{\text{igSPT}} U_{DW}^\dagger = -\sum_{i=1}^{L} \left( \sigma_i^x - \sigma_{i-1}^z \tau_{i-\frac{1}{2}}^x \sigma_i^x \tau_{i+\frac{1}{2}}^x \sigma_{i+1}^z + \tau_{i-\frac{1}{2}}^x \right). \tag{C.1}$$

Since the last term commutes with all other terms, the Hibert space can be divided into sectors with different $\tau^x$ configurations. In different sectors, the sign of term $\sigma_{i-1}^z \sigma_i^x \sigma_{i+1}^z$ is decided

by $\tau^x_{i-\frac{1}{2}}\tau^x_{i+\frac{1}{2}}$. It is easy to see that the number of terms with $\tau^x_{i-\frac{1}{2}}\tau^x_{i+\frac{1}{2}} = -1$ must be even, since $\prod_{i=1}^{L} \tau^x_{i-\frac{1}{2}}\tau^x_{i+\frac{1}{2}} = 1$. We prove the splitting of ground state energy of first two terms in (C.1) with different $\tau$ configuration is order of $1/L$ or exactly zero. Therefore, when $L$ is large enough, the state in the ground state sector of (C.1) satisfies $\tau^x_{i+\frac{1}{2}} = 1$ for each $i$.

When $L \in 2\mathbb{Z}+1$, we can prove the first two terms in (C.1) with any $\tau$ configuration can be mapped to the standard Levin-Gu model by a unitary transformation.

This implies the ground state energy of any $\tau$ configuration is same as that of the standard Levin-Gu model. To see the unitary transformation, let us assume that the sign of two terms $\sigma^z_{i-1}\sigma^x_i\sigma^z_{i+1}$ and $\sigma^z_{j-1}\sigma^x_j\sigma^z_{j+1}$ are both $-1$ where $1 \le i < j \le L$.[21] There is always a unitary transformation which can cancel these two $-1$ and preserve sign of other terms: If $i,j$ are both odd (even), the unitary transformation is $\prod_{i<2k<j} \sigma^x_{2k}$ ($\prod_{i<2k+1<j} \sigma^x_{2k+1}$). If $i$ is odd (even) and $j$ is even (odd), the unitary transformation is $\prod_{i<2k<L} \sigma^x_{2k}\prod_{1\le 2k+1<j}\sigma^x_{2k+1}$ ($\prod_{j<2k<L} \sigma^x_{2k}$ $\prod_{1\le 2k+1<i}\sigma^x_{2k+1}$) which can do the job only when $L \in 2\mathbb{Z}+1$. Since the number of terms with $-1$ sign is even, we can cancel these $-1$s step by step and obtain the standard Levin-Gu model at last.

When $L \in 2\mathbb{Z}$, we apply the unitary transformation (B.14) and (B.16) on the first two terms and then obtain XX chain with several minus coupling constants:

$$H_{\mu^1,\mu^2} = -\sum_{j=1}^{L}(\mu^1_{j,j+1}\sigma^z_j\sigma^z_{j+1} + \mu^2_{j,j+1}\sigma^x_j\sigma^x_{j+1}), \tag{C.2}$$

where $\mu^1$ and $\mu^2$ can be $\pm 1$. They are decided by the configuration of $\tau^x$ but we don't need to know the exact relationship. We only use the fact that $l + l' \in 2\mathbb{Z}$ where $l$ and $l'$ are number of $-1$ in $\mu^1$ and $\mu^2$.[22]

We note that the spectrum of Hamiltonian (C.2) only depends on $l, l'$ mod 2, and is independent of the configuration of $\mu^1$ and $\mu^2$. The reason is as follows. The sites of $-1$ in $\mu^1$ can be labeled as $\mu^1_{j_1,j_1+1}, \mu^1_{j_2,j_2+1}, \cdots \mu^1_{j_l,j_l+1}$ where $j_1 < j_2 < \cdots < j_l$. After the unitary transformation $\prod_{k=j_i+1}^{j_{i+1}} \sigma^x_k$, $\mu_{j_i,j_i+1}$ , $\mu_{j_{i+1},j_{i+1}+1}$ will become 1 without changing spectrum. Similar for $\mu^2$.

As $l + l'$ are even, there are only two equivalence classes for spectrum: $l = l' = 0$ and $l = l' = 1$. The first case is XX chain with PBC. In the second case, we can choose $\mu^1_{L,1} = \mu^2_{L,1} = -1$ without loss of generality. This is XX chain with the ABC. The splitting between ground state energy of these two boundary conditions is order of $1/L$ which completes our proof.

Besides, one can apply this argument to the $\mathbb{Z}_4$ igSPT with TBC and OBC as well. Generally, the ground state sector is Hilbert subspace which has eigenvalue 1 of the third term in the Hamiltonian (45), (47) and (50).

# D Edge Degeneracy of gSPT and igSPT

In section 2.3.3 and 3.2.3, we discussed the degeneracy of gSPT and igSPT under OBC by studying the dimension of irreducible representation of operators commuting with the Hamiltonian. In this appendix, we rederive the degeneracy under OBC in an alternative way. We first undecorate the domain wall which maps the gSPT and igSPT to the Ising and Levin-Gu models under OBC respectively, and then use the results in appendix B to rederive the degeneracy.

---

[21] We only focus on the "fundamental domain" where $1 \le i < j \le L$ and do not use periodicity $i \sim i+L$ here.

[22] $l + l' \in 2\mathbb{Z}$ can be seen from the transformation (B.14) and (B.16), which maps $\sigma^x_{2j-1} \to \sigma^z_{2j-1}\sigma^z_{2j}$, $\sigma^x_{2j} \to \sigma^x_{2j-1}\sigma^x_{2j}$, $-\sigma^z_{2j-1}\sigma^x_{2j}\sigma^z_{2j+1} \to \sigma^x_{2j}\sigma^x_{2j+1}$ and $-\sigma^z_{2j}\sigma^x_{2j+1}\sigma^z_{2j+2} \to \sigma^z_{2j}\sigma^z_{2j+1}$.

### D.1 Edge Degeneracy of $\mathbb{Z}_2 \times \mathbb{Z}_2$ gSPT

In section 2.3.3, we studied the $\mathbb{Z}_2 \times \mathbb{Z}_2$ gSPT under OBC, with the Hamiltonian (22),

$$H_{\text{gSPT}}^{\text{OBC}} = -\sum_{i=1}^{L-1}\left(\sigma_i^z \tau_{i+\frac{1}{2}}^x \sigma_{i+1}^z + \sigma_i^z \sigma_{i+1}^z\right) - \sum_{i=2}^{L} \tau_{i-\frac{1}{2}}^z \sigma_i^x \tau_{i+\frac{1}{2}}^z. \tag{D.1}$$

After $U_{DW}$ transformation, the Hamiltonian is given by

$$U_{DW} H_{\text{gSPT}}^{\text{OBC}} U_{DW}^\dagger = -\sum_{i=1}^{L-1}\left(\tau_{i+\frac{1}{2}}^x + \sigma_i^z \sigma_{i+1}^z\right) - \sum_{i=2}^{L} \sigma_i^x. \tag{D.2}$$

$\tau_{L+\frac{1}{2}}$ decouples from the Hamiltonian which gives two ground state degeneracy. The $\sigma_1^z$ commutes with Hamiltonian which gives two fixed boundary conditions on the left end and the right end is free boundary condition. Therefore we have four exact ground states. But this is unstable under symmetric perturbations as noted in section 2.3.3. We can add the boundary term (25) which becomes

$$-\sigma_L^z \tau_{L+\frac{1}{2}}^x, \tag{D.3}$$

after conjugated by $U_{DW}$, i.e. domain wall undecoration. Now $\tau_{L+\frac{1}{2}}^x$ no longer decouples, which lifts degeneracy due to free boundary condition on the right, and ground state degeneracy reduces to two.

### D.2 Edge degeneracy of $\mathbb{Z}_4$ igSPT

In section 3.2.3, we studied the $\mathbb{Z}_4$ igSPT under OBC, with the Hamiltonian (50)

$$H_{\text{igSPT}}^{\text{OBC}} = -\sum_{i=2}^{L}\left(\tau_{i-\frac{1}{2}}^z \sigma_i^x \tau_{i+\frac{1}{2}}^z + \tau_{i-\frac{1}{2}}^y \sigma_i^x \tau_{i+\frac{1}{2}}^y\right) - \sum_{i=1}^{L-1} \sigma_i^z \tau_{i+\frac{1}{2}}^x \sigma_{i+1}^z. \tag{D.4}$$

After undecorating the domain wall, we obtain the Levin-Gu model under OBC

$$U_{DW} H_{\text{igSPT}}^{\text{OBC}} U_{DW}^\dagger = -\sum_{i=1}^{L-1} \tau_{i+\frac{1}{2}}^x - \sum_{i=2}^{L-1}\left(\sigma_i^x - \sigma_{i-1}^z \tau_{i-\frac{1}{2}}^x \sigma_i^x \tau_{i+\frac{1}{2}}^x \sigma_{i+1}^z\right) - \left(\sigma_L^x - \sigma_{L-1}^z \tau_{L-\frac{1}{2}}^x \sigma_L^x \tau_{L+\frac{1}{2}}^x\right). \tag{D.5}$$

The ground state should be the eigenstate of $\tau_{i-\frac{1}{2}}^x$ ($i < L+1$) with eigenvalue 1. The low energy effective Hamiltonian is:

$$U_{DW} H_{\text{igSPT}}^{\text{OBC}} U_{DW}^\dagger \Big|_{\text{low}} = -\sum_{i=2}^{L-1}\left(\sigma_i^x - \sigma_{i-1}^z \sigma_i^x \sigma_{i+1}^z\right) - \left(\sigma_L^x - \sigma_{L-1}^z \sigma_L^x \tau_{L+\frac{1}{2}}^x\right). \tag{D.6}$$

Since $\tau_{L+\frac{1}{2}}^x$ commute with effective Hamiltonian, we can redefine $\tau_{L+\frac{1}{2}}^x$ as $\sigma_{L+1}^z$ and (D.6) becomes (B.24) with system size $L + 1$. We thus conclude that when $L \in 2\mathbb{Z}+1$, the ground state degeneracy is four and when $L \in 2\mathbb{Z}$ the ground state degeneracy is two.

## E $\mathbb{Z}_4^{\mathbb{T}} \times \mathbb{Z}_2$ igSPT

In this section we discuss another example of igSPT which respects the $\mathbb{Z}_4^{\mathbb{T}} \times \mathbb{Z}_2$ symmetries. We will also discuss the PBC, TBC and OBC.

### E.1 Lattice Hamiltonian

Let us assign three spin-$\frac{1}{2}$s $\tau, \sigma$ and $\mu$ per unit cell and the Hamiltonian is:

$$H_{\mathbb{Z}_4^{\mathbb{T}} \times \mathbb{Z}_2} = \sum_j \left( \mu_j^z \tau_{j+\frac{1}{2}}^x \mu_{j+1}^z + \sigma_j^z \mu_j^z \tau_{j+\frac{1}{2}}^x \mu_{j+1}^z \sigma_{j+1}^z + \sigma_j^x \mu_j^x + \sigma_j^x \right) - \sum_j \tau_{j-\frac{1}{2}}^z \mu_j^x \tau_{j+\frac{1}{2}}^z. \tag{E.1}$$

This Hamiltonian respects the following symmetry:

$$\mathbb{Z}_4^{\mathbb{T}} : U_{\mathbb{T}} \equiv \prod_j \left( \frac{1+\mu_j^x}{2} \sigma_j^x + \frac{1-\mu_j^x}{2} i\sigma_j^y \right) K, \qquad U_{\mathbb{T}}^2 = \prod_j \mu_j^x, \tag{E.2}$$

$$\mathbb{Z}_2^{\tau} : U_{\tau} \equiv \prod_j \tau_j^x, \tag{E.3}$$

where $\mathbb{T}$ stands for time reversal, and $K$ is the complex conjugation.

To see that (E.1) is a $\mathbb{Z}_2 \times \mathbb{Z}_4^{\mathbb{T}}$ igSPT, we show that it can be obtained by starting with a $\mathbb{Z}_2^{\tau} \times \mathbb{Z}_2^{\mathbb{T}}$ anomalous critical theory, and decorating the $\mathbb{Z}_2^{\tau}$ domain wall by 1d $\mathbb{Z}_2^{\mu}$ gapped SPT, where $\mathbb{Z}_2^{\mu}$ is generated by $U_{\mathbb{T}}^2$. Let us apply $U_{DW}$ of $\tau$ and $\mu$ on both the Hamiltonian (E.1) and the symmetry operators (E.2) and (E.3).

$$U_{DW} U_{\mathbb{T}} U_{DW}^{\dagger} = \prod_j \left( \frac{1 + \mu_j^x \tau_{j-\frac{1}{2}}^z \tau_{j+\frac{1}{2}}^z}{2} \sigma_j^x + \frac{1 - \mu_j^x \tau_{j-\frac{1}{2}}^z \tau_{j+\frac{1}{2}}^z}{2} i\sigma_j^y \right) K, \tag{E.4}$$

$$U_{DW} U_{\tau} U_{DW}^{\dagger} = U_{\tau}, \tag{E.5}$$

$$U_{DW} H_{\mathbb{Z}_4^{\mathbb{T}} \times \mathbb{Z}_2} U_{DW}^{\dagger} = \sum_j (\tau_{j+\frac{1}{2}}^x + \sigma_j^z \tau_{j+\frac{1}{2}}^x \sigma_{j+1}^z + \sigma_j^x + \tau_{j-\frac{1}{2}}^z \sigma_j^x \mu_j^x \tau_{j+\frac{1}{2}}^z) - \sum_j \mu_j^x. \tag{E.6}$$

In (E.6), since the last term commutes with all other terms, the energy eigenstates are eigenstates of $\mu_j^x$. Similar to the proof in the $\mathbb{Z}_4$ igSPT, we can consider the spectrum of first four terms in the Hamiltonian (E.6) with different configurations of $\mu^x$. These four terms can be mapped to an XX chain by applying the unitary transformations (B.14):

$$H(\{\mu_j^x\}) = \sum_{j=1}^{L} \sigma_j^z \tau_{j+\frac{1}{2}}^z + \tau_{j+\frac{1}{2}}^z \sigma_{j+1}^z + \sigma_j^x \tau_{j+\frac{1}{2}}^x + \tau_{j-\frac{1}{2}}^x \sigma_j^x \mu_j^x. \tag{E.7}$$

According to the proof in appendix C, we know the spectrum of the (E.7) is invariant if we flip even number of $\mu^x$. Thus, the spectrum of first four terms in (E.6) is that of XX chain with boundary condition: $\sigma_{L+j}^x = \pm \sigma_j^x$ and $\sigma_{L+j}^z = \sigma_j^z$, where we take $\pm$ sign if there are even or odd number of $\mu^x = -1$ respectively. After taking the continuum limit (B.19), these two boundary conditions are PBC and ABC for $\theta$ respectively. The splitting between the corresponding ground state energy is also of order $1/L$. Thus in the low energy state sector, one can find that $\mu_j^x = 1$. The effective Hamiltonian and symmetry are those of the boundary model of 2+1d $\mathbb{Z}_2^{\mathbb{T}} \times \mathbb{Z}_2$ SPT [7]:

$$U_{DW} U_{\mathbb{T}} U_{DW}^{\dagger} \Big|_{\text{low}} = \prod_j \left( \frac{1 + \tau_{j-\frac{1}{2}}^z \tau_{j+\frac{1}{2}}^z}{2} \sigma_j^x + \frac{1 - \tau_{j-\frac{1}{2}}^z \tau_{j+\frac{1}{2}}^z}{2} i\sigma_j^y \right) K, \tag{E.8}$$

$$U_{DW} H_{\mathbb{Z}_4^{\mathbb{T}} \times \mathbb{Z}_2} U_{DW}^{\dagger} \Big|_{\text{low}} = \sum_j \left( \tau_{j+\frac{1}{2}}^x + \sigma_j^z \tau_{j+\frac{1}{2}}^x \sigma_{j+1}^z + \sigma_j^x + \tau_{j-\frac{1}{2}}^z \sigma_j^x \tau_{j+\frac{1}{2}}^z \right). \tag{E.9}$$

Moreover the proof on the equivalence between ground state sector and XX chain can be generalized to twisted boundary conditions and open boundary conditions. We conclude that the ground state sector of different boundary conditions is always Hibert subspace which has eigenvalue 1 of the last term in the Hamiltonian (E.10) and (E.14).

### E.2 Charge of twisted boundary condition

We show that the charge of the ground state under TBC is nontrivial, implying that (E.1) is a nontrivial igSPT. Let us start by twisting the boundary condition using the $\mathbb{Z}_2^\tau$ symmetry, which we denote as $\mathbb{Z}_2^\tau$-TBC. The Hamiltonian (E.1) becomes

$$
H_{\mathbb{Z}_4^{\mathbb{T}}\times\mathbb{Z}_2}^{\mathbb{Z}_2^\tau} = \sum_{j=1}^{L}\left(\mu_j^z\tau_{j+\frac{1}{2}}^x\mu_{j+1}^z + \sigma_j^z\mu_j^z\tau_{j+\frac{1}{2}}^x\mu_{j+1}^z\sigma_{j+1}^z + \sigma_j^x\mu_j^x + \sigma_j^x\right)
$$
$$
-\left(\sum_{j=1}^{L-1}\tau_{j-\frac{1}{2}}^z\mu_j^x\tau_{j+\frac{1}{2}}^z - \tau_{L-\frac{1}{2}}^z\mu_j^x\tau_{\frac{1}{2}}^z\right).
$$

The ground state satisfies

$$
\tau_{j-\frac{1}{2}}^z\mu_j^x\tau_{j\frac{1}{2}}^z = 1 \qquad (0 < j < L), \qquad \tau_{L-\frac{1}{2}}^z\mu_j^x\tau_{\frac{1}{2}}^z = -1. \tag{E.10}
$$

which implies that the ground state has a nontrivial $\mathbb{Z}_2^\mu$ charge

$$
\prod_{j=1}^{L}\mu_j^x \left|\text{GS}\right\rangle_{\text{tw}}^{\mathbb{Z}_2^\tau} = -\left|\text{GS}\right\rangle_{\text{tw}}^{\mathbb{Z}_2^\tau}. \tag{E.11}
$$

On the other hand, if we twist by $\mathbb{Z}_2^\mu$ symmetry, the SPT criticality Hamiltonian becomes

$$
H_{\mathbb{Z}_4^{\mathbb{T}}\times\mathbb{Z}_2}^{\mathbb{Z}_2^\mu} = \sum_{j=1}^{L-1}\left(\mu_j^z\tau_{j+\frac{1}{2}}^x\mu_{j+1}^z + \sigma_j^z\mu_j^z\tau_{j+\frac{1}{2}}^x\mu_{j+1}^z\sigma_{j+1}^z\right) + \sum_{j=1}^{L}\left(\sigma_j^x\mu_j^x + \sigma_j^x - \tau_{j-\frac{1}{2}}^z\mu_j^x\tau_{j+\frac{1}{2}}^z\right)
$$
$$
-\mu_L^z\tau_{\frac{1}{2}}^x\mu_1^z - \sigma_L^z\mu_L^z\tau_{\frac{1}{2}}^x\mu_1^z\sigma_1^z
$$
$$
= \tau_{\frac{1}{2}}^z H_{\mathbb{Z}_4^{\mathbb{T}}\times\mathbb{Z}_2}\tau_{\frac{1}{2}}^z. \tag{E.12}
$$

It is straightforward to check that $\left|\text{GS}\right\rangle_{\text{tw}}^{\mathbb{Z}_2^\mu}$ has $\mathbb{Z}_2^\tau$ charge 1:

$$
U_\tau\left|\text{GS}\right\rangle_{\text{tw}}^{\mathbb{Z}_2^\mu} = U_\tau\tau_{\frac{1}{2}}^z U_\tau^\dagger U_\tau\left|\text{GS}\right\rangle = -\tau_{\frac{1}{2}}^z\left|\text{GS}\right\rangle = -\left|\text{GS}\right\rangle_{\text{tw}}^{\mathbb{Z}_2^\mu}. \tag{E.13}
$$

### E.3 Open boundary condition

To consider OBC, we truncate the spin chain so that $\sigma$-spins and $\mu$-spins live on $i = 1,\dots,L$, and $\tau$-spins live on $i = \frac{3}{2},\dots,L+\frac{1}{2}$. We only keep the terms in (E.1) that are fully supported on the spin chain. The Hamiltonian is

$$
H_{\mathbb{Z}_4^{\mathbb{T}}\times\mathbb{Z}_2}^{\text{OBC}} = \sum_{j=1}^{L-1}\mu_j^z\tau_{j+\frac{1}{2}}^x\mu_{j+1}^z + \sigma_j^z\mu_j^z\tau_{j+\frac{1}{2}}^x\mu_{j+1}^z\sigma_{j+1}^z + \sum_{j=1}^{L}\sigma_j^x\mu_j^x + \sigma_j^x - \sum_{j=2}^{L}\tau_{j-\frac{1}{2}}^z\mu_j^x\tau_{j+\frac{1}{2}}^z. \tag{E.14}
$$

There are two boundary operators $\mu_1^x\tau_{\frac{3}{2}}^z$ and $\tau_{L+\frac{1}{2}}^z$ commuting with Hamiltonian. Since both of them anticommute with $U_\tau$, there must be at least two exactly degenerate ground states of (E.14).

The exact ground state degeneracy can be determined by undecorating the domain wall, by applying $U_{DW}$ on (E.14):

$$
U_{DW}H_{\mathbb{Z}_4^{\mathbb{T}}\times\mathbb{Z}_2}^{\text{OBC}}U_{DW}^\dagger = \sum_{j=1}^{L-1}\tau_{j+\frac{1}{2}}^x + \sigma_j^z\tau_{j+\frac{1}{2}}^x\sigma_{j+1}^z + \sum_{j=1}^{L}\sigma_j^x + \sum_{j=2}^{L}\tau_{j-\frac{1}{2}}^z\sigma_j^x\mu_j^x\tau_{j+\frac{1}{2}}^z + \sigma_1^x\mu_1^x\tau_{\frac{3}{2}}^z - \sum_{j=2}^{L}\mu_j^x,
$$

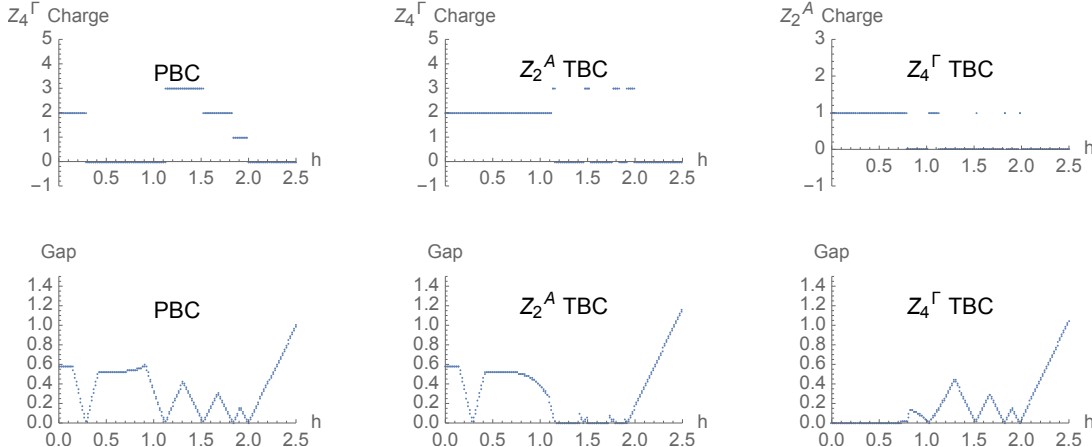

Figure 5: $\mathbb{Z}_4^\Gamma$ charge of the ground state under PBC, relative $\mathbb{Z}_4^\Gamma$ charge of the ground state under $\mathbb{Z}_2^A$ TBC, relative $\mathbb{Z}_2^A$ charges of the ground state under $\mathbb{Z}_4^\Gamma$ TBC, and the gap between the ground state and first excited state under PBC and two TBC's. The horizontal axis is the perturbation strength (52). The system size is $L = 11$.

and the two boundary operators becomes $\mu_1^x$ and $\tau_{L+\frac{1}{2}}^z$. In the ground state sector $\mu_j^x = 1$ for $2 \leq j \leq L$. The Hamiltonian in the low energy then simplifies to

$$U_{DW} H_{\mathbb{Z}_4^{\mathbb{T}} \times \mathbb{Z}_2}^{\mathrm{OBC}} U_{DW}^\dagger \Big|_{\mathrm{low}} = \sum_{j=1}^{L-1} (\tau_{j+\frac{1}{2}}^x + \sigma_j^z \tau_{j+\frac{1}{2}}^x \sigma_{j+1}^z) + \sum_{j=1}^L \sigma_j^x + \sum_{j=2}^L \tau_{j-\frac{1}{2}}^z \sigma_j^x \tau_{j+\frac{1}{2}}^z + \sigma_1^x \mu_1^x \tau_{\frac{3}{2}}^z . \quad \text{(E.15)}$$

Under the unitary transformation (B.14), this Hamiltonian is mapped to

$$U \left( U_{DW} H_{\mathbb{Z}_4^{\mathbb{T}} \times \mathbb{Z}_2}^{\mathrm{OBC}} U_{DW}^\dagger \Big|_{\mathrm{low}} \right) U^\dagger = \sum_{j=1}^{L-1} \sigma_j^z \tau_{j+\frac{1}{2}}^z + \tau_{j+\frac{1}{2}}^z \sigma_{j+1}^z + \sum_{j=1}^L \sigma_j^x \tau_{j+\frac{1}{2}}^x + \sum_{j=2}^L \tau_{j-\frac{1}{2}}^x \sigma_j^x + \sigma_1^x \mu_1^x , \quad \text{(E.16)}$$

and the two boundary operators become $\mu_1^x$ and $\tau_{L+\frac{1}{2}}^x$. The Hamiltonian (E.16) can be understood as an XX chain on an open chain with size $2L$ and one spin-$\frac{1}{2}$ per unit cell.

Similar to the $\mathbb{Z}_4$ igSPT, we can redefine $\mu_1^x$ as $\tau_{\frac{1}{2}}^x$. After taking the continuum limit (B.19), $\sigma^x$ and $\tau^x$ are mapped to $\sin \theta$. Thus $\mu_1^x = \pm 1$ and $\tau_{L+\frac{1}{2}}^x = \pm 1$ correspond to the boundary conditions $\sin \theta(x = 0/L) = \pm 1$ which implies $\theta(x = 0/L) = \pm \frac{\pi}{2}$. There is an energy splitting between the ground states of two boundary conditions

$$E_{(\frac{\pi}{2}, -\frac{\pi}{2})/(-\frac{\pi}{2}, \frac{\pi}{2})} - E_{(\frac{\pi}{2}, \frac{\pi}{2})/(-\frac{\pi}{2}, -\frac{\pi}{2})} \propto \frac{1}{L} . \quad \text{(E.17)}$$

In summary, the ground state degeneracy under OBC is two.

# F Small-scale numerical study for igSPT under perturbation

In this appendix, we perform the exact diagonalization numerically, and record the lowest $h$ where the charges jump in table 4. We also plot the charges and the gaps under various boundary conditions for $L = 11$ (22 spin-$\frac{1}{2}$'s) in figure 5.

From the plots in figure 5, we find that the $\mathbb{Z}_4^\Gamma$ charge under PBC and both relative charges under TBC's are unchanged until $h$ reaches the first critical value $h_c \simeq 0.28$. This first transition is probed by the charge jump under PBC, where the finite size gap closes simultaneously.

Table 4: Lowest $h$ where the symmetry charge of the ground state under three boundary conditions jumps, for $L = 4, 5, 7, 8, 9, 11$.

| $L$ | $\mathbb{Z}_4^\Gamma$ Charge under PBC | $\mathbb{Z}_4^\Gamma$ Charge under $\mathbb{Z}_2^A$-TBC | $\mathbb{Z}_2^A$ Charge under $\mathbb{Z}_4^\Gamma$-TBC |
|---|---|---|---|
| 4 | 1.01 | 1.01 | 1.01 |
| 5 | 1.30 | 1.30 | 0.50 |
| 7 | 0.44 | 1.32 | 0.98 |
| 8 | 0.70 | 0.70 | 0.70 |
| 9 | 0.86 | 0.86 | 0.86 |
| 11 | 0.28 | 1.12 | 1.01 |

When $h$ further passes $h_c$, the system goes through a sequence of transitions, some are probed by the $\mathbb{Z}_2^A$-TBC, some are probed by the $\mathbb{Z}_4^\Gamma$-TBC and the others are probed by PBC. When $h$ is sufficiently large ($h > 2$), the system enters into a trivially gapped phase, and all charges become trivial, which is consistent with the phase diagram by the Kennedy-Tasaki transformation in [33].

For different system sizes, for instance $L = 5$ as shown in table 4, the first transition can be probed by the relative charge under TBC instead. Hence it is important to examine all the boundary conditions and find the minimal $h_c$ where the charge jumps. We plot the minimal $h_c$ for each $L$ in figure 6.

The above discussion seems to suggest that igSPT is more stable than the gSPT. Let us however make a cautionary remark. As observed in figure 6, the critical perturbation strength $h_c$ depends on the system size $L$. At this point with small-scale ED study, we are unable to conclude whether the transition away from the igSPT at $L \to \infty$ happens at immediately after $h = 0$ or at a finite $h_c$.

However, the analytical result by the Kennedy-Tasaki transformation shows that where $h_c$ converges to a finite value in the thermodynamical limit [33]. It would also be interesting to study more sophisticated perturbation than (52) which can drive the system to the trivially gapped phase, and discuss the transition for small perturbation strength.

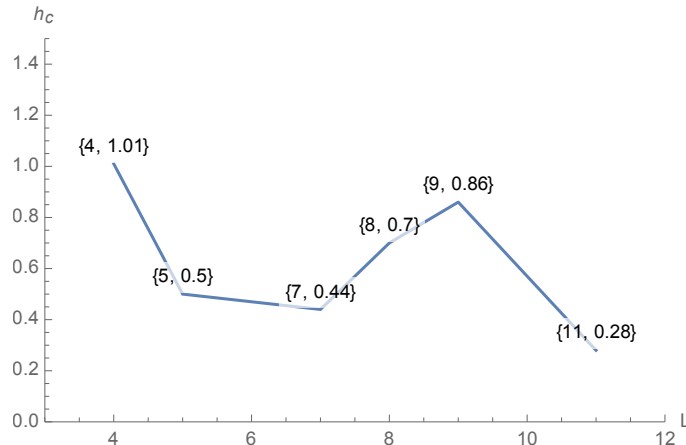

Figure 6: System size $L$ dependence of the first transition out of the $\mathbb{Z}_4^\Gamma$ igSPT.

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
