# Peer review of "Decorated Defect Construction of Gapless-SPT States"

_SciPost Physics, doi:SciPost Phys. 17, 013 (2024)_

## Round 1 · Referee Report · Anonymous · 2023-11-22

Report

I thank the authors for carefully considering many of the concerns I raised in my first report. The manuscript has been considerably updated as a result. To highlight a few notable changes:

* The authors have overhauled Sec 2.4 where they study the instability of gSPT. Unlike their previous model, I now agree with their claims about the results of perturbing this model.

* The authors have rewritten part of their work to better reflect the pre-existing understanding in the literature. This includes referring to Ref. 14 [Scaffidi et al] for the gSPT example, to Ref. 15 [Verresen et al] for the relation between string orders and twisted boundary conditions, and to Ref. 16 [Thorngren et al] for a framework for understanding igSPTs. This helps with framing the novel additions of the present work.

* The authors have stepped away from the 'SPTC' terminology and have highlighted that the key part of the present work is about how decorated domain walls are relevant for constructing both gSPTs and igSPTs.

* The authors have restructured the discussion of their igSPT example to make manifest how decorated domain walls are used to construct this example, rather than retro-actively analyze the known model.

These changes have certainly improved the manuscript. I think the focus on decorated domain walls is a good one: although this is a known mechanism for constructing gSPTs (as now accurately pointed out by the authors), to the best of my knowledge it had not yet been used to construct igSPT. This seems to be echoed by another referee's report ("Report 1"). I quite enjoyed the (updated) discussion in Sec. 3.1.2 where the authors start with the Z2 anomalous symmetry and use domain wall decoration to give a new derivation of the igSPT.

For these reasons, I would like to recommend publication of this manuscript in one of SciPost's journals. However, I notice two inaccuracies in the current manuscript, so my recommendation would be conditional on correcting these:

1) The authors imply that non-triviality of a gSPT (or igSPT) can be determined by seeing a non-trivial charge in twisted boundary conditions (TBC). This is misleading: one needs to separately argue/check that this charge assignment is meaningful and robust. While the authors check that the TBC has a unique g.s., this does not necessarily mean it gives a robust invariant, since exploring a moduli space of CFTs might cause a level-crossing in the TBC which toggles these symmetry assignments. In Sec. 2 this possibility can be excluded since the Ising CFT has no moduli space, however Sec. 3 requires an additional discussion since the compact boson CFT considered there does admit a marginal perturbation. Hence, to conclude that one has a non-trivial (i)gSPT, one has to argue that the charge of the TBC cannot change without a non-analytic change of the CFT. Examples of such 'washing out' of invariants in compact boson CFTs have been discussed in, e.g., Ref 15 (Sec V.C). This shows that claiming a non-trivial gapless SPT requires a study of the actual CFT properties, since certain CFTs (and their moduli spaces) can absorb/remove the properties of certain domain wall decorations.

2) Although the (in)stability study in Sec. 2 has been addressed, the (in)stability study in Sec. 3 (for the igSPT) remains problematic. The ED study for L<=11 (which means <=5-6 sites for the gapless sublattice) gives little to no information about the system at hand. Firstly, the extracted values for h jump extremely for the sizes considered (Fig 5), which means there is virtually no meaningful information to be extracted from this. Secondly, just checking the symmetry charges does not tell whether one remains in the gapless state or whether one has perturbed e.g. into a nearby symmetry-breaking state. Correspondingly, the two logical options given on p26 present a false dichotomy; it does not even allow for the likely option of perturbing the igSPT into a symmetry-breaking state. This actually happens if one perturbs with \sum_n (-1)^n tau^x instead of \sum_n tau^x : then the igSPT immediately ("h_c = 0") flows to a symmetry-breaking state, in violation of the supposed dichotomy presented by the authors. In short, since there is virtually no meaningful information to be extracted from the current ED results, I worry that including/discussing it can only lead to confusion. I encourage the authors to simply omit it, or to move it to an appendix, or at the very least tone down the surrounding discussion even more (by removing the false dichotomy).

Conditional on resolving these two remaining issues, I would support publication of this manuscript in one of SciPost's journals. I am undecided about whether the manuscript meets the criteria of "SciPost Physics", but it does of "SciPost Physics Core". I say this because all the models discussed in the present work are already discussed in other works (in particular, the model in Sec 2 appeared in Ref 14, Sec 3 in Ref 23 [up to Jordan-Wigner], and Sec 4 in Ref 61), and Ref 23 already offered a unifying perspective of gSPT and igSPT in terms of SPT-like cocycles. Nevertheless, there is great value in the lattice-based examples of decorated domain walls presented in this work, and I believe this can make the concepts of previous works more readily accessible to a broader audience.

In addition to the above two important remarks, I have a few smaller comments which the authors can choose to consider:

* In the discussion of igSPT in Sec. 1.3.2, the authors say "Concretely, the symmetry breaking phase we started with has a particular anomaly of a particular (non-normal) subgroup \tilde \Gamma of \Gamma". I believe this is incorrect: \tilde \Gamma is a quotient group, not a subgroup

* Below Eq. 3.12, the authors say "Since U is on-site, Z4 now is anomaly free", which is misleading, since U was already anomaly-free as soon as it was extended to Z4. The definition of anomaly-free does not require a unitary to be (manifestly) on-site.

* Although there is footnote 8 which says that certain models of this work are related to certain models in Ref. 44, this is quite non-explicit. It might be clearer to mention somewhere in Sec 3 that Eq. 3.11 is Jordan-Wigner dual to Eqs.49&50 in Ref. 44?

  • validity: -
  • significance: -
  • originality: -
  • clarity: -
  • formatting: -
  • grammar: -

Author:  Yunqin Zheng  on 2024-04-13  [id 4418]

(in reply to Report 1 on 2023-11-22)

We thank the referee for the illuminating comments. Please find our response below.

The first comment is very interesting. The referee pointed out a scenario where the charge in the twisted sector can jump as the CFT moves along the moduli space. One example were this can happen is the CFT at the transition between a trivial phase and a $G$ gapped SPT phases (in Ref 16, the authors used the transition between tivial and Haldane phase as an example, note that this is not the type of phase transition (i.e. trivial-SPT transition) in the current draft). At the transition, there is an additional symmetry ($\mathbb{Z}_2$ symmetry in the above example) that has a mixed anomaly with the $G$ symmetry. Deforming the original CFT to another CFT through the CFT with an accidental symmetry with $G$ mixed anomaly can indeed change the ground state symmetry properties we discussed in the paper. Assuming this is the only mechanism (See arXiv: 2312.16898 for a recent discussion), then we need to restrict the marginal deformation to the one such that such accidental symmetry (with mixed anomaly with $G$) does not emerge upon deformation, which indeed requires detailed study of the CFT and deformation. We have commented this in footnote 16 of the updated draft.

For the second comment, we basically agree with the referee. But we would like to mention that our ED study for $L=11$ means 11 unit cells (with 22 spin-$\frac12$'s). We have also shortened the discussion of stability of igSPT according to referee's suggestion, and demoted the discussion to an appendix F. We also deleted the dichotomy.

We have also addressed the remaining smaller comments listed at the end of the referee report.

---

## Round 1 · Author Response

This version is a major revision of the previous version.

---

## Editorial Decision

published